# Provably Robust Detection of Out-of-distribution Data (almost) for free

## Abstract

The application of machine learning in safety-critical systems requires a reliable assessment of uncertainty. However, deep neural networks are known to produce highly overconfident predictions on out-of-distribution (OOD) data. Even if trained to be non-confident on OOD data one can still adversarially manipulate OOD data so that the classifier again assigns high confidence to the manipulated samples. In this paper we propose a novel method that combines a certifiable OOD detector with a standard classifier from first principles into an OOD aware classifier. This way we achieve the best of two worlds: certifiably adversarially robust OOD detection, even for OOD samples close to the in-distribution, without loss in either prediction accuracy or detection performance for non-manipulated OOD data. Moreover, due to the particular construction our classifier provably avoids the asymptotic overconfidence problem of standard neural networks.

## 1 Introduction

Deep neural networks have achieved state-of-the-art performance in many application domains. However, the widespread usage of deep neural networks in safety-critical applications, e.g. in healthcare, autonomous driving/aviation, manufacturing, raises concerns as deep neural networks have problematic deficiencies. Among these deficiencies are overconfident predictions on non-task related inputs (Nguyen et al., 2015; Hendrycks & Gimpel, 2017)which has recently attracted a lot of interest. ReLU networks have even been shown to be provably overconfident far away from the training data (Hein et al., 2019). However, reliable confidences of the classifier on the classification task (in-distribution) (Guo et al., 2017) as well as on the out-distribution (Hendrycks & Gimpel, 2017; Hein et al., 2019) are important to be able to detect when the deep neural network is working outside of its specification, which can then be used to either involve a human operator or to fall back into a "safe state". Thus, solving this problem is of high importance for trustworthy ML systems. Many approaches have been proposed for OOD detection, (Hendrycks & Gimpel, 2017; Liang et al., 2018; Lee et al., 2018a;b; Hendrycks et al., 2019; Ren et al., 2019; Hein et al., 2019; Meinke & Hein, 2020; Chen et al., 2020; Papadopoulos et al., 2021; Macêdo & Ludermir, 2021; Macêdo et al., 2021) and one of the currently best performing methods enforces low confidence during training ("outlier exposure" (OE)) on a large and diverse set of out-distribution images (Hendrycks et al., 2019) which leads to strong separation of in- and out-distribution based on the confidence of the classifier. Crucially, this also generalizes to novel test out-distributions.

However, current OOD detection methods are vulnerable to adversarial manipulations, i.e. small adversarial modifications of OOD inputs lead to large confidence of the classifier on the manipulated samples (Nguyen et al., 2015; Hein et al., 2019; Sehwag et al., 2019). While different methods for adversarially robust OOD detection have been proposed (Hein et al., 2019; Sehwag et al., 2019; Meinke & Hein, 2020; Chen et al., 2020; Bitterwolf et al., 2020) there is little work on *provably adversarially* robust OOD detection (Meinke & Hein, 2020; Bitterwolf et al., 2020; Kopetzki et al., 2020; Berrada et al., 2021). In CCU (Meinke & Hein, 2020) they append density estimators based on Gaussian mixture models for in- and out-distribution to the softmax layer. By also enforcing low confidence on a training out-distribution, they achieve similar OOD detection performance to (Hendrycks et al., 2019) but can guarantee that the classifier shows decreasing confidence as one moves away from the training data. However, for close in-distribution inputs this approach yields no guarantee as the Gaussian mixture models are not powerful enough for complex image classification

Table 1: **ProoD combines desirable properties of existing (adversarially robust) OOD detection methods.** It has high test accuracy and standard OOD detection performance (as (Hendrycks et al., 2019)) and has worst-case guarantees if the out-distribution samples are adversarially perturbed in an $l_\infty$-neighborhood to maximize the confidence (see Section 4.2). Similar to CCU (Meinke & Hein, 2020) it avoids the problem of asymptotic overconfidence far away from the training data.

| | OE | CCU | ACET/ATOM | GOOD | ProoD |
|---|---|---|---|---|---|
| High accuracy | ✓ | ✓ | ✓ | | ✓ |
| High clean OOD detection performance | ✓ | ✓ | ✓ | | ✓ |
| Adv. OOD $l_\infty$-robustness | | | (✓) | ✓ | ✓ |
| Adv. OOD $l_\infty$-certificates | | | | ✓ | ✓ |
| Provably not asympt. overconfident | | ✓ | | | ✓ |

tasks. In (Kristiadi et al., 2020a;b) similar asymptotic guarantees are derived for Bayesian neural networks but without any robustness guarantees. In (Kopetzki et al., 2020) they apply randomized smoothing to obtain guarantees wrt. $l_2$-perturbations for Dirchlet-based models (Malinin & Gales, 2018; 2019; Sensoy et al., 2018) which already show quite some gap in terms of AUC-ROC to SOTA OOD detection methods even without attacks. Interval bound propagation (IBP) (Gowal et al., 2018; Mirman et al., 2018; Zhang et al., 2020; Jovanović et al., 2021) has been shown to be one of the most effective techniques in certified adversarial robustness on the in-distribution when applied during training. In GOOD (Bitterwolf et al., 2020) they use IBP to compute upper bounds on the confidence in an $l_\infty$-neighborhood of the input and minimize these upper bounds on a training out-distribution. This yields classifiers with pointwise guarantees for adversarially robust OOD detection even for "close" out-distribution inputs which generalize to novel OOD test distributions. However, the employed architectures of the neural network are restricted to rather shallow networks as otherwise the bounds of IBP are loose. Thus, they obtain low classification accuracy which is far from the state-of-the-art, e.g. 91% on CIFAR-10, and their approach does not scale to more complex tasks like ImageNet. In particular, despite its low accuracy the employed network architecture is quite large and has higher memory consumption than a ResNet50. Moreover, one does not get any guarantees on the asymptotic behavior of the classifier as one moves away from the training data. The authors of (Berrada et al., 2021) use SOTA verification techniques (Dathathri et al., 2020) and get guarantees for OOD detection wrt. $l_\infty$-perturbations for ACET models (Hein et al., 2019) that were not specifically trained to be verifiable but the guarantees obtained by training the models via IBP in (Bitterwolf et al., 2020) are significantly better.

In this paper we propose ProoD which merges a certified binary discriminator for in-versus out-distribution with a classifier for the in-distribution task in a principled fashion into a joint classifier. This combines the advantages of CCU (Meinke & Hein, 2020) and GOOD (Bitterwolf et al., 2020) without suffering from their downsides. In particular, ProoD simultaneously achieves the following:

- Guaranteed adversarially robust OOD detection via confidence upper bounds on $l_\infty$-balls around OOD samples.

- Additionally, it provably prevents the asymptotic overconfidence of deep neural networks.

- It can be used with arbitrary architectures and has no loss in prediction performance and standard OOD detection performance.

Thus, we get provable guarantees for adversarially robust OOD detection, fix the asymptotic over-confidence (almost) for free as we have (almost) no loss in prediction and standard OOD detection performance. We qualitatively compare the properties of our model to prior approaches in Table 1.

## 2 PROVABLY ROBUST DETECTION OF OUT-OF-DISTRIBUTION DATA

In the following we consider feedforward networks for classification, $f : \mathbb{R}^d \to \mathbb{R}^K$, with $K$ classes defined with $x^{(0)} = x$ as

$$x^{(l)} = \sigma^{(l)} \left( W^{(l)} x^{(l-1)} + b^{(l)} \right) \quad l = 1, \dots L - 1, \qquad f(x) = W^{(L)} x^{(L-1)} + b^{(L)},$$

$$(1)$$

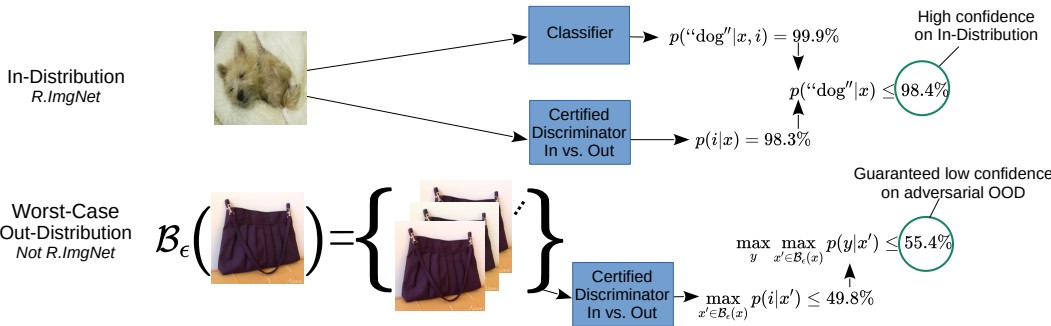

Figure 1: **ProoD's Architecture:** Our method produces high confidence on the in-distribution sample (R.ImgNet) using Eq. (2). It also provides an upper bound on the worst-case confidence over all images within a neighborhood of the shown OOD sample (OpenImages) using Eq. (3). Bounding the confidence on this infinite set of OOD samples is possible without any loss in accuracy.

where $L \in \mathbb{N}$ is the number of layers, $W^{(l)}$ and $b^{(l)}$ are weights and biases and $\sigma^{(l)}$ is either the ReLU or leaky ReLU activation function of layer $l$. We refer to the output of $f$ as the *logits* and get a probability distribution over the classes via $\hat{p}(y|x) = \frac{e^{f_y(x)}}{\sum_k^K e^{f_k(x)}}$ for $y = 1, \ldots, K$. We define the confidence as $\mathrm{Conf}(f(x)) = \max_{y=1,\ldots,K} \hat{p}(y|x)$.

## 2.1 JOINT MODEL FOR OOD DETECTION AND CLASSIFICATION

In our joint model we assume that there exists an in- and out-distribution where the out-distribution samples are unrelated to the in-distribution task. Thus, we can formally write the conditional distribution on the input as

$$\hat{p}(y|x) = \hat{p}(y|x,i)\hat{p}(i|x) + \hat{p}(y|x,o)\hat{p}(o|x), \tag{2}$$

where $\hat{p}(i|x)$ is the conditional distribution that sample $x$ belongs to the in-distribution and $\hat{p}(y|x,i)$ is the conditional distribution for the in-distribution. We assume that OOD samples are unrelated and thus maximally un-informative to the in-distribution task, i.e. we fix $\hat{p}(y|x,o) = \frac{1}{K}$. In (Meinke & Hein, 2020) they further decomposed $\hat{p}(i|x) = \frac{\hat{p}(x|i)\hat{p}(i)}{\hat{p}(x)}$ and used Gaussian mixture models to estimate $\hat{p}(x|i)$ with fixed $\hat{p}(i) = \hat{p}(o) = \frac{1}{2}$. As the properties of the Gaussian mixture model can be directly controlled in comparison to a more powerful generative model such as VAEs, this allowed (Meinke & Hein, 2020) to prove that $\hat{p}(y|x)$ becomes uniform over the classes if $x$ is far away from the training data. However, the downside of the Gaussian mixture models is that this approach yields no guarantees for close out-distribution samples. Instead in this paper we directly learn $\hat{p}(i|x)$ which results in a binary classification problem and we train this binary classifier in a certified robust fashion wrt. an $l_\infty$-threat model so that even adversarially manipulated OOD samples are detected. In order to avoid confusion with the multi-class classifier, we will refer to $\hat{p}(i|x)$ as a binary discriminator. In an $l_\infty$-ball of radius $\epsilon$ around $x \in \mathbb{R}^d$ and for all $y$ we have the upper bound (also see App. H)

$$\max_{\|x'-x\|_\infty \le \epsilon} \hat{p}(y|x') \le \max_{\|x'-x\|_\infty \le \epsilon} \hat{p}(i|x') + \frac{1}{K}\big(1 - \hat{p}(i|x')\big) = \frac{K-1}{K} \max_{\|x'-x\|_\infty \le \epsilon} \hat{p}(i|x') + \frac{1}{K}, \tag{3}$$

so we can defer the certification "work" to the binary discriminator. Using a particular constraint on the weights of the binary discriminator, we get similar asymptotic properties as in (Meinke & Hein, 2020) but additionally get certified adversarial robustness for close out-distribution samples as in (Bitterwolf et al., 2020). In contrast to (Bitterwolf et al., 2020) this comes without loss in test accuracy or non-adversarial OOD detection performance as in our model the neural network used for the in-distribution classification task $\hat{p}(y|x,i)$ is independent of the binary discriminator. Thus, we have the advantage that the classifier can use arbitrary deep neural networks and is not constrained to certifiable networks. We call our approach **Pr**ovable **o**ut-**o**f-**D**istribution detector (ProoD) and visualize its components in Figure 1.

**Certifiably Robust Binary Discrimination of In- versus Out-Distribution** The first goal is to get a certifiably adversarially robust OOD detector $\hat{p}(i|x)$. We train this binary discriminator independent

of the overall classifier as the training schedules for certified robustness are incompatible with the standard training schedules of normal classifiers. For this binary classification problem we use a logistic model $\hat{p}(i|x) = \frac{1}{1+e^{-g(x)}}$, where $g : \mathbb{R}^d \to \mathbb{R}$ are logits of a neural network (we denote the weights and biases of $g$ by $W_g$ and $b_g$ in order to discriminate it from the classifier $f$ introduced in the next paragraph). Let $(x_r, y_r)_{r=1}^N$ be our in-distribution training data (we use the class encoding $+1$ for the in-distribution and $-1$ for out-distribution) and $(z_s)_{s=1}^M$ be our training out-distribution data. Then the optimization problem associated to the binary classification problem becomes:

$$\min_{\substack{g \\ W_g^{(L_g)}<0}} \frac{1}{N} \sum_{r=1}^N \log\left(1 + e^{-g(x_r)}\right) + \frac{1}{M} \sum_{s=1}^M \log\left(1 + e^{\bar{g}(z_s)}\right), \quad (4)$$

where we minimize over the parameters of the neural network $g$ under the constraint that the weights of the output layer $W_g^{(L_g)}$ are componentwise negative and $\bar{g}(z) \geq \max_{u \in B_p(z,\epsilon)} g(u)$ is an upper bound on the output of $g$ around OOD samples for a given $l_p$-threat model $B_p(z,\epsilon) = \{u \in [0,1]^d \mid \|u - z\|_p \leq \epsilon\}$. In this paper we always use an $l_\infty$-threat model. This upper bound could, in principle, be computed using any certification technique but we will use interval bound propagation (IBP) since it is simple, fast and has been shown to produce SOTA results (Gowal et al., 2018). Note that this is not standard adversarial training for a binary classification problem as here we have an asymmetric situation: we want to be (certifiably) robust to adversarial manipulation on the out-distribution data but *not* on the in-distribution and thus the upper bound is only used for out-distribution samples. The negativity of the output layer's weights $W_g^{(L_g)}$ is enforced by using the parameterization $(W_g^{(L_g)})_j = -e^{h_j}$ componentwise and optimizing over $h_j$. In Section 3 we show how the negativity of $W_g^{(L_g)}$ allows us to control the asymptotic behavior of the joint classifier.

For the reader's convenience we quickly present the upper $\overline{x}^{(l)}$ and lower $\underline{x}^{(l)}$ bounds on the output of layer $l$ in a feedforward neural network produced by IBP:

$$\overline{x}^{(l)} = \sigma\left(W_+^{(l)}\overline{x}^{(l-1)} + W_-^{(l)}\underline{x}^{(l-1)} + b^{(l)}\right), \quad \underline{x}^{(l)} = \sigma\left(W_+^{(l)}\underline{x}^{(l-1)} + W_-^{(l)}\overline{x}^{(l-1)} + b^{(l)}\right), \quad (5)$$

where $W_+ = \max(0, W)$ and $W_- = \min(0, W)$ (min/max used componentwise). For an $l_\infty$-threat model one starts with the upper and lower bounds for the input layer $\overline{x}^{(0)} = x + \epsilon$ and $\underline{x}^{(0)} = x - \epsilon$ and then iteratively computes the layerwise upper and lower bounds $\overline{x}^{(l)}, \underline{x}^{(l)}$ which fulfill

$$\underline{x}^{(l)} \leq \min_{\|x'-x\|_\infty \leq \epsilon} x^{(l)}(x') \leq \max_{\|x'-x\|_\infty \leq \epsilon} x^{(l)}(x') \leq \overline{x}^{(l)}. \quad (6)$$

While in (Bitterwolf et al., 2020) they also used IBP to upper bound the confidence of the classifier this resulted in a bound that took into account all $\mathcal{O}(K^2)$ logit differences between all classes. In contrast, our loss in Eq. (4) is significantly simpler as we just have a binary classification problem and therefore only need a single bound. Thus, our approach easily scales to tasks with a large number of classes and training the binary discriminator with IBP turns out to be significantly more stable than the approach in (Bitterwolf et al., 2020) and does not require many additional tricks.

**(Semi)-Joint Training of the final Classifier**   Given the certifiably robust model $\hat{p}(i|x)$ for the binary classification task between in- and out-distribution, we need to determine the final predictive distribution $\hat{p}(y|x)$ in Eq. (2). On top of the provable OOD performance that we get from Eq. (3), we also want to achieve SOTA performance on unperturbed OOD data. In principle we could independently train a model for the predictive in-distribution task $\hat{p}(y|x, i)$, e.g. using outlier exposure (OE) (Hendrycks et al., 2019) or any other state-of-the-art OOD detection method and simply combine it with our $\hat{p}(i|x)$. While this does lead to models with high OOD performance that also have guarantees, it completely ignores the interaction between $\hat{p}(i|x)$ and $\hat{p}(y|x, i)$ during training. Instead we propose to train $\hat{p}(y|x, i)$ by optimizing our final predictive distribution $\hat{p}(y|x)$. Note that in order to retain the guarantees of $\hat{p}(i|x)$ we only train the parameters of the neural network $f : \mathbb{R}^d \to \mathbb{R}^K$ and need to keep $\hat{p}(i|x)$ resp. $g$ fixed. Because $g$ stays fixed we call this semi-joint training. We use OE (Hendrycks et al., 2019) for training $\hat{p}(y|x)$ with the cross-entropy loss and use

the softmax-function in order to obtain the predictive distribution $\hat{p}_f(y|x, i) = \frac{e^{f_y(x)}}{\sum_k e^{f_k(x)}}$ from $f$:

$$\min_f -\frac{1}{N} \sum_{r=1}^{N} \log\left(\hat{p}(y_r|x_r)\right) - \frac{1}{M} \sum_{s=1}^{M} \frac{1}{K} \sum_{l=1}^{K} \log\left(\hat{p}(l|z_s)\right)$$

$$= \min_f -\frac{1}{N} \sum_{r=1}^{N} \log\left(\hat{p}_f(y_r|x_r, i)\hat{p}(i|x_r) + \frac{1}{K}\left(1 - \hat{p}(i|x_r)\right)\right)$$

$$- \frac{1}{M} \sum_{s=1}^{M} \frac{1}{K} \sum_{l=1}^{K} \log\left(\hat{p}_f(l|z_s, i)\hat{p}(i|z_s) + \frac{1}{K}\left(1 - \hat{p}(i|z_s)\right)\right), \tag{7}$$

where the first term is the standard cross-entropy loss on the in-distribution but now for our joint model for $\hat{p}(y|x)$ and the second term enforces uniform confidence on out-distribution samples. In App. B we show that semi-joint training leads to stronger guarantees than separate training.

The loss in Eq. (4) implicitly weighs the in-distribution and worst-case out-distribution equally, which amounts to the assumption $p(i) = \frac{1}{2} = p(o)$. This highly conservative choice simplifies training the binary discriminator but may not reflect the expected frequency of OOD samples at test time and in effect means that $\hat{p}(i|x)$ tends to be quite low. This typically yields good guaranteed AUCs but can have a negative impact on the standard out-distribution performance. In order to better explore the trade-off of guaranteed and standard OOD detection, we repeat the above semi-joint training with different shifts of the offset parameter in the output layer

$$b' = b_g^{(L_g)} + \Delta, \tag{8}$$

where $\Delta \geq 0$ leads to increasing $\hat{p}(i|x)$. This shift has a direct interpretation in terms of the probabilities $p(i)$ and $p(o)$. Under the assumption that our binary discriminator $g$ is perfect, that is

$$p(i|x) = \frac{p(x|i)p(i)}{p(x|i)p(i) + p(x|o)p(o)} = \frac{1}{1 + \frac{p(x|o)p(o)}{p(x|i)p(i)}} = \frac{1}{1 + e^{-g(x)}}, \tag{9}$$

then it holds that $e^{g(x)} = \frac{p(x|i)p(i)}{p(x|o)p(o)}$. A change of the prior probabilities $\tilde{p}(i)$ and $\tilde{p}(o)$ without changing $p(x|i)$ and $p(x|o)$ then corresponds to a novel classifier

$$e^{\tilde{g}(x)} = \frac{p(x|i)\tilde{p}(i)}{p(x|o)\tilde{p}(o)} = \frac{p(x|i)p(i)}{p(x|o)p(o)} \frac{p(o)\tilde{p}(i)}{p(i)\tilde{p}(o)} = e^{g(x)}e^{\Delta}, \quad \text{with} \quad \Delta = \log\left(\frac{p(o)\tilde{p}(i)}{p(i)\tilde{p}(o)}\right). \tag{10}$$

Note that $\tilde{p}(i) > p(i)$ corresponds to positive shifts. In a practical setting, this parameter can be chosen based on the priors for the particular application. Since no such priors are available in our case we determine a suitable shift by evaluating on the training out-distribution, see Section 4.2 for details. Please note that we explicitly do not train the shift parameter since this way the guarantees would get lost as the classifier implicitly learns a large $\Delta$ in order to maximize the confidence on the in-distribution, thus converging to a normal outlier exposure-type classifier without any guarantees.

## 3 GUARANTEES ON ASYMPTOTIC CONFIDENCE

In this section we show that our specific construction provably avoids the issue of asymptotic overconfidence that was pointed out in (Hein et al., 2019). Note that the resulting guarantee (as stated in Theorem 1) is different from and in addition to the robustness guarantees discussed in the previous section (see Eq. (3)). The previous section dealt with providing confidence upper bounds on neighborhoods around OOD samples whereas this section deals with ensuring that a classifier's confidence decreases asymptotically as one moves away from all training data.

We note that a ReLU neural network $f : \mathbb{R}^d \to \mathbb{R}^K$ as defined in Eq. (1) using ReLU or leaky ReLU as activation functions, potential max-or average pooling and skip connection yields a piece-wise affine function (Arora et al., 2018; Hein et al., 2019), i.e. there exists a finite set of polytopes $Q_r \subset \mathbb{R}^d$ with $r = 1, \ldots, R$ such that $\cup_{r=1}^{R} Q_r = \mathbb{R}^d$ and $f$ restricted to each of the polytopes is an affine function. Since there are only finitely many polytopes some of them have to extend to infinity and on these ones the neural network is essentially an affine classifier. This fact has been used in (Hein et al.,

2019) to show that ReLU networks are almost always asymptotically overconfident in the sense that if one moves to infinity the confidence of the classifier approaches 1 (instead of converging to $1/K$ as in these regions the classifier has never seen any data). The following result taken from (Hein et al., 2019) basically says that as one moves to infinity by upscaling a vector one eventually ends up in a polytope which extends to infinity.

**Lemma 1** ((Hein et al., 2019)). *Let $\{Q_r\}_{r=1}^R$ be the set of convex polytopes on which a ReLU-network $f : \mathbb{R}^d \to \mathbb{R}^K$ is an affine function, that is for every $k \in \{1, \dots, R\}$ and $x \in Q_k$ there exists $V^k \in \mathbb{R}^{K \times d}$ and $c^k \in \mathbb{R}^K$ such that $f(x) = V^k x + c^k$. For any $x \in \mathbb{R}^d$ with $x \neq 0$ there exists $\alpha \in \mathbb{R}$ and $t \in \{1, \dots, R\}$ such that $\beta x \in Q_t$ for all $\beta \geq \alpha$.*

The following theorem now shows that, opposite to standard ReLU networks (Hein et al., 2019), our proposed joint classifier gets provably less confident in its decisions as one moves away from the training data which is a desired property of any reasonable classifier.

**Theorem 1.** *Let $x \in \mathbb{R}^d$ with $x \neq 0$ and let $g : \mathbb{R}^d \to \mathbb{R}$ be the ReLU-network of the binary discriminator and denote by $\{Q_r\}_{r=1}^R$ the finite set of polytopes such that $g$ is affine on these polytopes which exists by Lemma 1. Denote by $Q_t$ the polytope such that $\beta x \in Q_t$ for all $\beta \geq \alpha$ and let $x^{(L-1)}(z) = Uz + d$ with $U \in \mathbb{R}^{n_{L-1} \times d}$ and $d \in \mathbb{R}^{n_{L-1}}$ be the output of the pre-logit layer of $g$ for $z \in Q_t$. If $Ux \neq 0$, then*

$$\lim_{\beta \to \infty} \hat{p}(y|\beta x) = \frac{1}{K}.$$

The proof is in App. C. In App. A we see that the condition $Ux \neq 0$ is not restrictive, as this property holds in all cases where we checked it for our joint classifier. The negativity condition on the weights $W_g^{(L_g)}$ of the output layer of the in-vs. out-distribution discriminator $g$ is crucial for the proof. This condition may seem restrictive, but we did not encounter any negative influence of this constraint on test accuracy, guaranteed or standard OOD detection performance. Thus the asymptotic guarantees come essentially for free. In (Meinke & Hein, 2020) they derived non-asymptotic guarantees and it would be relatively easy to also achieve this for the joint classifier via a decay factor for $\hat{p}(i|x)$ that depends on the distance to the training data but we prefer not to enforce this explicitly.

## 4 EXPERIMENTS

### 4.1 TRAINING OF PROOD

We provide experiments on CIFAR10, CIFAR100 (Krizhevsky & Hinton, 2009) and Restricted Imagenet (R.ImgNet) (Tsipras et al., 2018). The latter consists of images from the ILSVRC2012 subset of ImageNet (Deng et al., 2009; Russakovsky et al., 2015) belonging to 9 types of animals.

**Training the Binary Discriminator**   We train the binary discriminator between in-and out-distribution using the loss in Eq. (4) with the bounds over an $l_\infty$-ball of radius $\epsilon = 0.01$ for the out-distribution following (Bitterwolf et al., 2020). We use relatively shallow CNNs with only 5 layers plus pooling layers, see App. D. For the training out-distribution, we could follow previous work and use 80M Tiny Images (Torralba et al., 2008) for CIFAR10 and CIFAR100. However, there have been concerns over the use of this dataset (Birhane & Prabhu, 2021) because of offensive class labels. Although we do not use any of the class labels, we choose to use OpenImages (Kuznetsova et al., 2020) as training OOD instead. In order to ensure a fair comparison with prior work we also present results that were obtained using 80M Tiny Images in App. E. For R.ImgNet we use the ILSVRC2012 train images that do not belong to R.ImgNet as training out-distribution (NotR.ImgNet).

**Semi-Joint Training**   For the classifier we use a ResNet18 architecture on CIFAR and a ResNet50 on R.ImgNet. Note that the architecture of our binary discriminator is over an order of magnitude smaller than the one in (Bitterwolf et al., 2020) (11MB instead of 135MB) and thus the memory overhead for the binary discriminator is less than a third of that of the classifier. All schedules, hardware and hyperparameters are described in App. D. As discussed in Section 2.1 when training the binary discriminator one implicitly assumes that in- and (worst-case) out-distribution samples are equally likely. It seems very unlikely that one would be presented with such a large number of OOD samples in practice but as discussed in Section 2.1, we can adjust the weight of the losses after training

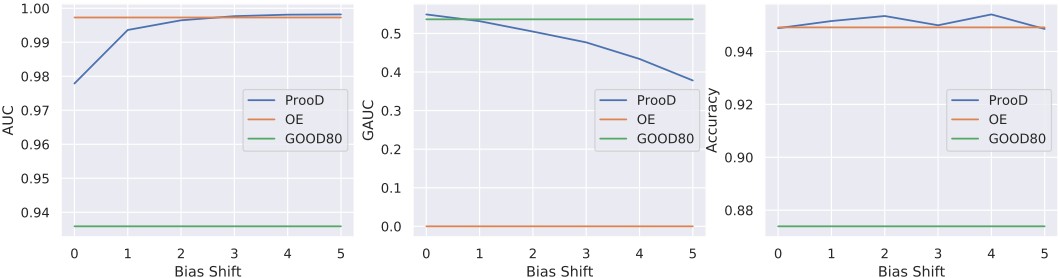

Figure 2: **(Provable) OOD performance depends on the bias:** Using CIFAR10 as the in-distribution and the test set of OpenImages as OOD we plot the test accuracy, AUC and GAUC (see Section 4.2) as a function of the bias shift $\Delta$ (see Eq. (8)). For small $\Delta$ the AUCs tend to be worse than for the OE model, but small bias shifts also provide stronger guarantees so some trade-off exists.

the discriminator (but before training the classifier) by shifting the bias $b_g^{(L_g)}$ in the output layer of the binary discriminator. We train several ProoD models for binary shifts in $\{0, 1, 2, 3, 4, 5, 6\}$ and then evaluate the AUC and guaranteed AUC (see 4.2) on a subset of the training out-distribution OpenImages (resp. NotR.ImgNet). For all bias shifts we use the same fixed provably trained binary discriminator and only train the classifier part. As our goal is to have provable guarantees with minimal or no loss on the standard OOD detection task, among all solutions which have better AUC than outlier exposure (OE) (Hendrycks et al., 2019) we choose the one with the highest guaranteed AUC on OpenImages (on CIFAR10/CIFAR100) resp. NotR.ImgNet (on R.ImgNet). If none of the solutions has better AUC than OE on the training out-distribution we take the one with the highest AUC. We show the trade-off curves for the example of CIFAR10 in Figure 2. The corresponding figures for CIFAR100 and R.ImgNet can be found in App. D.

## 4.2 EVALUATION

**Setup** For OOD evaluation for CIFAR10/100 we use the test sets from CIFAR100/10, SVHN (Netzer et al., 2011), the classroom category of downscaled LSUN (Yu et al., 2015) (LSUN_CR) as well as smooth noise as suggested in (Hein et al., 2019) and described in App. D. For R.ImgNet we use Flowers (Nilsback & Zisserman, 2008), FGVC Aircraft (Maji et al., 2013), Stanford Cars (Krause et al., 2013) and smooth noise as test out-distributions. Since the computation of adversarial AUCs (next paragraph) requires computationally expensive adversarial attacks, we restrict the evaluation on the out-distribution to a fixed subset of 1000 images (300 in the case of LSUN_CR) for the CIFAR experiments and 400 for the R.ImgNet models. We still use the entire test set for the in-distribution.

**Guaranteed and Adversarial AUC** We use the confidence of the classifier as the feature to discriminate between in- and out-distribution samples. While in standard OOD detection one uses the area under the receiver-operator characteristic (AUC) to measure discrimination of in- from out-distribution, Bitterwolf et al. (2020) introduced the worst-case AUC (WCAUC) which is defined as the minimal AUC one can achieve if all out-distribution samples are allowed to be perturbed to reach maximal confidence within a certain threat model, which in our case is an $l_\infty$-ball of radius $\epsilon$. The AUC and WAUC of a feature $h : \mathbb{R}^d \to \mathbb{R}$ is defined as:

$$\text{AUC}_h(p_1, p_2) = \mathop{\mathbb{E}}_{\substack{x \sim p_1 \\ z \sim p_2}} \left[ \mathbb{1}_{h(x) > h(z)} \right], \quad \text{WCAUC}_h(p_1, p_2) = \mathop{\mathbb{E}}_{\substack{x \sim p_1 \\ z \sim p_2}} \left[ \mathbb{1}_{h(x) > \max_{\|z' - z\|_\infty \le \epsilon} h(z')} \right], \quad (11)$$

where $p_1, p_2$ are in-resp. out-distribution and with slight abuse of notation the indicator function $\mathbb{1}$ returns 1 if the expression in its argument is true and 0 otherwise. For all but one of our baselines, the OOD detecting feature $h$ is the confidence of the classifier. Since the exact evaluation of the WCAUC is computationally infeasible, we compute an upper bound and lower bound on the WCAUC by finding $\underline{h}(z) \le \max_{\|z' - z\|_\infty \le \epsilon} h(z') \le \bar{h}(z)$. We find the upper bound on the WCAUC, the adversarial AUC (AAUC), by maximizing the confidence using an adversarial attack inside the $l_\infty$-ball (i.e. finding an $\underline{h}$) and we compute a lower bound on the WCAUC, the guaranteed AUC (GAUC), by using upper on the confidence inside the $l_\infty$-ball via IBP (i.e. $\bar{h}$). For non-provable methods, no non-trivial

upper bound $\bar{h} < \infty$ is available so their GAUCs are always 0. Note that our threat model is different from adversarial robustness on the in-distribution which neither our method nor the baselines pursue.

Gradient obfuscation (Papernot et al., 2017; Athalye et al., 2018) poses a significant challenge for the evaluation of AAUCs (Bitterwolf et al., 2020) so we employ an ensemble of different versions of projected gradient descent (PGD) (Madry et al., 2018) as well as SquareAttack (Andriushchenko et al., 2020) with 5000 queries. We use APGD (Croce & Hein, 2020) (except on RImgNet, due to a memory leak) with 500 iterations and 5 random restarts. We also use a 200-step PGD attack with momentum of 0.9 and backtracking that starts with a step size of 0.1 which is halved every time a gradient step does not increase the confidence and gets multiplied by 1.1 otherwise. This PGD is applied to different starting points: i) a decontrasted version of the image, i.e. the point that minimizes the $l_\infty$-distance to the grey image $0.5 \cdot \vec{1}$ within the threat model, ii) 3 uniformly drawn samples from the threat model and iii) 3 versions of the original image perturbed by Gaussian noise with $\sigma = 10^{-4}$ and then clipped to the threat model. We always clip to the box $[0, 1]^d$ at each step of the attack. For all attacks and all models we directly optimize the final score that is used for OOD detection. Using different types of starting points is crucial for strong attacks on these OOD points, as some models have precisely zero gradient on many OOD samples.

**Baselines** We compare to a normally trained baseline (Plain) and outlier exposure (OE), both trained using the same architecture and hyperparameters as the classifier in ProoD. For both ATOM and ACET we found the models' OOD detection to be much less adversarially robust than claimed in (Chen et al., 2020) (see App. E) so we retrained their models using the same architecture, threat model and training out-distribution with their original code (for CIFAR10/100). Running these adversarial training procedures on ImageNet resolution is infeasibly expensive. For GOOD we also retrain using OpenImages as training OOD with the code from (Bitterwolf et al., 2020) (comparisons with their pre-trained models can be found in App. E). Since they are only available on CIFAR10, we attempted to train models on CIFAR100 using their code and the same hyperparameters and schedules as they used for CIFAR10. This only lead to models with accuracy below $25\%$, so we do not include these models in our evaluation. Since CCU was already shown to not provide benefits over OE on OOD data that is not very far from the in-distribution (e.g. uniform noise) (Meinke & Hein, 2020; Bitterwolf et al., 2020) we do not include it as a baseline. We also evaluate the OOD-performance of the provable binary discriminator (ProoD-Disc) that we trained for ProoD. Note that this is not a classifier and so it is included simply for reference. All results are shown in Table 2.

**Results** ProoD achieves non-trivial GAUCs on all datasets. As was also observed in (Bitterwolf et al., 2020), this shows that the IBP guarantees not only generalize to unseen samples but even to unseen distributions. In general the gap between our GAUCs and AAUCs is extremely small. This shows that the seemingly simple IBP bounds can be remarkably tight, as has been observed in other works (Gowal et al., 2018; Jovanović et al., 2021). It also shows that there would be very little benefit in applying stronger verification techniques like (Cheng et al., 2017; Katz et al., 2017; Dathathri et al., 2020) in ProoD. The bounds are also much tighter than for GOOD, which is likely due to the fact that for GOOD the confidence is much harder to optimize during an attack because it involves maximizing the confidence in an essentially random class.

For CIFAR10, on 3 out of 4 out-distributions ProoD's GAUCs are higher than ATOM's and ACET's AAUCs, i.e. our model's *provable* adversarial robustness exceeds the SOTA methods' *empirical adversarial* robustness in these cases. Note that this is *not* due to our retraining, because the authors' pre-trained models perform even more poorly (as shown in App. E). On CIFAR100 ProoD's guarantees are weaker and ATOM produces strong AAUCs. However, we observe that training both ACET and ATOM can produce inconsistent results, i.e. sometimes almost no robustness is achieved. For the successfully trained robust ATOM model on CIFAR100 we observe drastically reduced accuracy. Due to the difficulty in attacking these models, it is not unlikely that a more sophisticated attack could produce even lower AAUCs. Combined with the fact that both ACET and ATOM rely on expensive adversarial training procedures we argue that using ProoD is preferable in practice.

On CIFAR10 we see that ProoD's GAUCs are comparable to, if slightly worse than the ones of both $GOOD_{80}$ and $GOOD_{100}$. Note that although the presented GOOD models are retrained, the same observations hold true when comparing to the pre-trained models (see App. E). However, we want to point out that ProoD achieves this while retaining both high accuracy and OOD performance, both of which are lacking for GOOD. It is also noteworthy that the GOOD models' memory footprints

Table 2: **OOD performance:** For all models we report accuracy on the test set of the in-distribution and AUCs, guaranteed AUCs (GAUC), adversarial AUCs (AAUC) for different test out-distributions. The radius of the $l_\infty$-ball for the adversarial manipulations of the OOD data is $\epsilon = 0.01$ for all datasets. The bias shift $\Delta$ that was used for ProoD is shown for each in-distribution. The AAUCs and GAUCs for ProoD tend to be very close, indicating remarkably tight certification bounds.

| In: CIFAR10 | | CIFAR100 | | | SVHN | | | LSUN_CR | | | Smooth | | |
|---|---|---|---|---|---|---|---|---|---|---|---|---|---|
| | Acc | AUC | GAUC | AAUC | AUC | GAUC | AAUC | AUC | GAUC | AAUC | AUC | GAUC | AAUC |
| Plain | 95.01 | 90.0 | 0.0 | 0.7 | 93.8 | 0.0 | 0.3 | 93.1 | 0.0 | 0.5 | 98.0 | 0.0 | 0.7 |
| OE | 94.91 | 91.1 | 0.0 | 0.9 | 97.3 | 0.0 | 0.0 | 100.0 | 0.0 | 2.7 | 99.9 | 0.0 | 1.5 |
| ATOM | 93.63 | 78.3 | 0.0 | 21.7 | 94.4 | 0.0 | 24.1 | 79.8 | 0.0 | 20.1 | 99.5 | 0.0 | 73.2 |
| ACET | 93.43 | 86.0 | 0.0 | 4.0 | 99.3 | 0.0 | 4.6 | 89.2 | 0.0 | 3.7 | 99.9 | 0.0 | 40.2 |
| GOOD$_{80}$* | 87.39 | 76.7 | 47.1 | 57.1 | 90.8 | 43.4 | 76.8 | 97.4 | 70.6 | 93.6 | 96.2 | 72.9 | 89.9 |
| GOOD$_{100}$* | 86.96 | 67.8 | 48.1 | 49.7 | 62.6 | 34.9 | 36.3 | 84.9 | 74.6 | 75.6 | 87.0 | 76.1 | 78.1 |
| ProoD-Disc | - | 62.9 | 57.1 | 57.8 | 72.6 | 65.6 | 66.4 | 78.1 | 71.5 | 72.3 | 59.2 | 49.7 | 50.4 |
| ProoD $\Delta\!=\!3$ | 94.99 | 89.8 | 46.1 | 46.8 | 98.3 | 53.3 | 54.1 | 100.0 | 58.3 | 59.7 | 99.9 | 38.2 | 38.8 |

| In: CIFAR100 | | CIFAR10 | | | SVHN | | | LSUN_CR | | | Smooth | | |
|---|---|---|---|---|---|---|---|---|---|---|---|---|---|
| | Acc | AUC | GAUC | AAUC | AUC | GAUC | AAUC | AUC | GAUC | AAUC | AUC | GAUC | AAUC |
| Plain | 77.38 | 77.7 | 0.0 | 0.4 | 81.9 | 0.0 | 0.2 | 76.4 | 0.0 | 0.3 | 86.6 | 0.0 | 0.4 |
| OE | 77.25 | 77.4 | 0.0 | 0.2 | 92.3 | 0.0 | 0.0 | 100.0 | 0.0 | 0.7 | 99.5 | 0.0 | 0.5 |
| ATOM | 68.32 | 78.3 | 0.0 | 50.3 | 91.1 | 0.0 | 67.0 | 95.9 | 0.0 | 75.6 | 98.2 | 0.0 | 80.7 |
| ACET | 73.02 | 73.0 | 0.0 | 1.4 | 97.8 | 0.0 | 0.7 | 75.8 | 0.0 | 2.6 | 99.9 | 0.0 | 12.8 |
| ProoD-Disc | - | 56.1 | 52.1 | 52.3 | 61.0 | 58.2 | 58.4 | 70.4 | 66.9 | 67.1 | 29.6 | 26.4 | 26.5 |
| ProoD $\Delta\!=\!5$ | 77.16 | 76.6 | 17.3 | 17.4 | 91.5 | 19.7 | 19.8 | 100.0 | 22.5 | 23.1 | 98.9 | 9.0 | 9.0 |

| In: R.ImgNet | | Flowers | | | FGVC | | | Cars | | | Smooth | | |
|---|---|---|---|---|---|---|---|---|---|---|---|---|---|
| | Acc | AUC | GAUC | AAUC | AUC | GAUC | AAUC | AUC | GAUC | AAUC | AUC | GAUC | AAUC |
| Plain | 96.34 | 92.3 | 0.0 | 0.5 | 92.6 | 0.0 | 0.0 | 92.7 | 0.0 | 0.1 | 98.9 | 0.0 | 8.6 |
| OE | 97.10 | 96.9 | 0.0 | 0.2 | 99.7 | 0.0 | 0.4 | 99.9 | 0.0 | 1.8 | 98.0 | 0.0 | 1.9 |
| ProoD-Disc | - | 81.5 | 76.8 | 77.3 | 92.8 | 89.3 | 89.6 | 90.7 | 86.9 | 87.3 | 81.0 | 74.0 | 74.8 |
| ProoD $\Delta\!=\!4$ | 97.25 | 96.9 | 57.5 | 58.0 | 99.8 | 67.4 | 67.9 | 99.9 | 65.7 | 66.2 | 98.6 | 52.7 | 53.5 |

*Uses different architecture of classifier, see "Baselines" in Section 4.2.

are over twice as large as ProoD's. Generally, for ProoD the accuracy is comparable to OE and the OOD performance is similar or marginally worse. Thus ProoD shows that it is possible to achieve certifiable adversarial robustness on the out-distribution while keeping very good prediction and OOD detection performance. Note that all methods struggle on separating CIFAR10 and CIFAR100 when using OpenImages as training OOD (as compared to 80M Tiny Images in App. E).

To the best of our knowledge with R.ImgNet we provide the first worst case OOD guarantees on high-resolution images. The GAUCs are higher than on CIFAR which indicates that meaningful certification on higher resolution is more achievable on this task than one might expect. FGVC and Cars may seem simple to separate from the animals in R.ImgNet but the same cannot be said for Flowers which are difficult to provably distinguish from images of insects with flowers in them.

## 5 CONCLUSION

We have demonstrated how to combine a provably robust binary discriminator between in- and out-distribution with a standard classifier in order to simultaneously achieve high accuracy, high OOD detection performance as well as worst-case OOD guarantees that are comparable to previous works. Thus, we have combined the best properties of previous work with only a small increase in total model size and only a single hyperparameter. This suggests that certifiable adversarial robustness on the out-distribution (as opposed to the in-distribution) is indeed possible without losing accuracy. We further showed how in our model simply enforcing negativity in the final weights of the discriminator fixes the problem of asymptotic overconfidence in ReLU classifiers. Training ProoD models is simple and stable and thus ProoD provides OOD guarantees that come (almost) for free.

## REPRODUCIBILITY

All details necessary for reproducing our experiments are given in Section 4 and App. D, including hyperparameters and hyperparameter selection. In order to further aid reproducibility we include the source code in the supplemental material (github link in final version), together with instructions for how to use it. We will also provide pre-trained models for public download in the final version.

## ETHICS

The only potential ethical concern that we see in this work is our use of the retracted dataset 80M Tiny Images (Torralba et al., 2008). However, we clearly explain that the only reason we use the dataset is providing a fair comparison to prior work. We firmly believe that our work helps the community move away from the use of this dataset because we provide extensive experiments on a different training out-distribution, going as far as retraining several previous baselines on it.

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

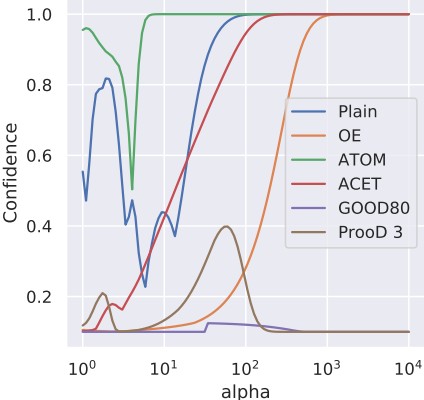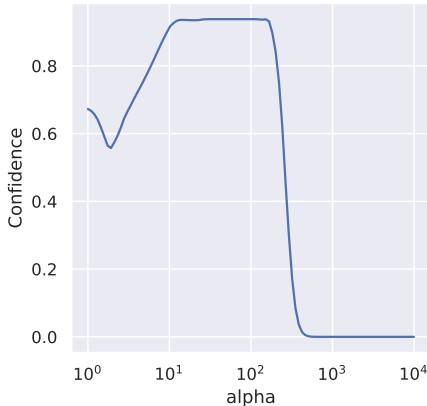

Figure 3: **Left, Asymptotic confidence:** We plot the mean confidence in the predicted in-distribution class for different models as one moves away from CIFAR100 samples along the trajectories $x + \alpha n$, where $n \in [-0.5, 0.5]^d$ and $\alpha \geq 0$. Only GOOD and ProoD converge to uniform confidence. **Right, Adversarial asymptotic confidence:** We try to find adversarial directions in which ProoD remains at a constant high confidence, as opposed to converging to low confidence. We plot the *maximum* of $\hat{p}(i|x)$ across 100 adversarially chosen directions as one moves further in these directions by factors of $\alpha$. Note that $\hat{p}(i|x) \to 0$ implies $\hat{p}(y|x) \to \frac{1}{K}$.

# A    ADVERSARIAL ASYMPTOTIC OVERCONFIDENCE

According to the authors of (Hein et al., 2019), under mild conditions, we should expect to find asymptotic overconfidence in all ReLU networks and almost all directions. In order to empirically evaluate this, we take different models that were trained on CIFAR10 and evaluate their confidence on different CIFAR100 samples. For each sample $x$ we track the confidence, $\max_k \hat{p}(k|x)$, along a trajectory in a uniform noise direction $x + \alpha n$, where $n \in [-0.5, 0.5]^d$ and $\alpha \geq 0$. The mean confidence across 100 such trajectories is shown on the left side of Figure 3. Even the models that produce low confidences on the original OOD sample asymptotically converge to maximal confidence far away. The only exceptions here are GOOD and ProoD and only ProoD can guarantee that the confidence cannot converge to 1.

However, even though the architecture provably prevents arbitrarily overconfident predictions and Theorem 1 ensures that most directions will indeed converge to uniform, it is, in principle, possible to find directions where the confidence $\hat{p}(i|x)$ remains constant if the condition $Ux \neq 0$ in Theorem 1 is not satisfied. We attempted to find such directions by running the following type of attack. We start from a random point $x \in [-0.5, 0.5]^d$ that we project onto a sphere of radius 100. We now run gradient descent (for 20000 steps), maximizing $g(x)$ while projecting onto the sphere at each step (unnormalized gradients with step size 0.1 for the first 10000 steps and 0.01 for the last 10000 steps). We then increase the radius to 1000 and run an additional 20000 steps with step size 0.1. We rescale the resulting direction vector down to an $l_\infty$-ball of norm 1 and compute the confidence $\hat{p}(i|x)$ as a function of the scaling in the adversarial directions. We show the resulting scale-wise *maximum* over 100 adversarial directions in Figure 3. Note that even the worst-case over 100 adversarially found directions decays to 0 asymptotically, thus empirically confirming the practical utility of Theorem 1. Note that the value of $\hat{p}(i|x)$ converging to 0 implies that the confidence of the ProoD model $\hat{p}(y|x)$ converges to 10%.

In Figure 3 GOOD also stands out as having low confidence in all directions that we studied. This is because in all the asymptotic regions that we looked at, the pre-activations of the penultimate layer are all negative. If one moves outward and these pre-activations only get more negative in all directions far away from the data, the confidence does, in fact, remain low. Unfortunately, it also leads to gradients that are precisely zero, which is why the same attack can not be applied here. However, there is no guarantee that GOOD does not also get in some direction asymptotically overconfident.

Table 3: **Architecture:** The architectures that are used for the binary discriminators. Each convolutional layer is directly followed by a ReLU.

| CIFAR | R.ImgNet |
|---|---|
| Conv2d(3, 128) | Conv2d(3, 128) |
| Conv2d(128, 256)$_{s=2}$ | AvgPool(2) |
| Conv2d(256, 256) | Conv2d(128, 256)$_{s=2}$ |
| AvgPool(2) | AvgPool(2) |
| FC(16384, 128) | Conv2d(256, 256) |
| FC(128, 1) | AvgPool(2) |
| | FC(50176, 128) |
| | FC(128, 1) |

## B SEPARATE TRAINING FOR PROOD

In Section 2.1 we describe semi-joint training of $\hat{p}(y|x)$. However, as pointed out in that section, it is possible to separately train a certifiable binary discriminator $\hat{p}(i|x)$ and an OOD aware classifier $\hat{p}(y|x, i)$ and to then simply combine them via Eq. (2). We call this method separate training ProoD-S and evaluate it by using an OE trained model for $\hat{p}(y|x, i)$. We show the results in Table 4, where we repeat the results for OE and ProoD for the reader's convenience. Note that OE and ProoD-S must always have the same accuracy on the in-distribution since they use the same model for classification (note that (2) preserves the ranking of $\hat{p}(y|x, i)$).

We see that the AUCs of ProoD-S are almost identical to those of OE. Without almost any loss in performance ProoD-S manages to provide non-trivial GAUCs. However, as one would expect, the semi-jointly trained ProoD provides stronger guarantees at similar clean performance. Nonetheless, this post-hoc method of adding some amount of certifiability to an existing system may be interesting in applications where retraining a deployed model from scratch is infeasible.

Table 4: **Separate Training:** Addendum to Table 2 showing the AUCs, GAUCs and AAUCs of ProoD-S on all datasets. The accuracy must always be identical to that of OE and the clean AUCs are also very similar to those of OE. The guarantees are strictly weaker than those provided by the semi-jointly trained ProoD.

| In: CIFAR10 | | CIFAR100 | | | SVHN | | | LSUN_CR | | | Smooth | | |
|---|---|---|---|---|---|---|---|---|---|---|---|---|---|
| | Acc | AUC | GAUC | AAUC | AUC | GAUC | AAUC | AUC | GAUC | AAUC | AUC | GAUC | AAUC |
| OE | 94.91 | 91.1 | 0.0 | 0.9 | 97.3 | 0.0 | 0.0 | 100.0 | 0.0 | 2.7 | 99.9 | 0.0 | 1.5 |
| ProoD-S $\Delta=3$ | 94.91 | 89.3 | 44.7 | 45.3 | 97.3 | 51.8 | 52.6 | 100.0 | 56.7 | 57.7 | 99.9 | 36.7 | 37.6 |
| ProoD $\Delta=3$ | 94.99 | 89.8 | 46.1 | 46.8 | 98.3 | 53.3 | 54.1 | 100.0 | 58.3 | 59.7 | 99.9 | 38.2 | 38.8 |

| In: CIFAR100 | | CIFAR10 | | | SVHN | | | LSUN_CR | | | Smooth | | |
|---|---|---|---|---|---|---|---|---|---|---|---|---|---|
| | Acc | AUC | GAUC | AAUC | AUC | GAUC | AAUC | AUC | GAUC | AAUC | AUC | GAUC | AAUC |
| OE | 77.25 | 77.4 | 0.0 | 0.2 | 92.3 | 0.0 | 0.0 | 100.0 | 0.0 | 0.7 | 99.5 | 0.0 | 0.5 |
| ProoD-S $\Delta=5$ | 77.25 | 77.4 | 17.2 | 17.3 | 92.3 | 19.5 | 19.6 | 100.0 | 22.4 | 22.6 | 99.5 | 9.0 | 9.1 |
| ProoD $\Delta=5$ | 77.16 | 76.6 | 17.3 | 17.4 | 91.5 | 19.7 | 19.8 | 100.0 | 22.5 | 23.1 | 98.9 | 9.0 | 9.0 |

| In: R.ImgNet | | Flowers | | | FGVC | | | Cars | | | Smooth | | |
|---|---|---|---|---|---|---|---|---|---|---|---|---|---|
| | Acc | AUC | GAUC | AAUC | AUC | GAUC | AAUC | AUC | GAUC | AAUC | AUC | GAUC | AAUC |
| OE | 97.10 | 96.9 | 0.0 | 0.2 | 99.7 | 0.0 | 0.4 | 99.9 | 0.0 | 1.8 | 98.0 | 0.0 | 1.9 |
| ProoD-S $\Delta=4$ | 97.10 | 96.9 | 50.1 | 50.7 | 99.7 | 59.7 | 60.6 | 99.9 | 57.9 | 58.9 | 98.0 | 40.8 | 42.3 |
| ProoD $\Delta=4$ | 97.25 | 96.9 | 57.5 | 58.0 | 99.8 | 67.4 | 67.9 | 99.9 | 65.7 | 66.2 | 98.6 | 52.7 | 53.5 |

## C PROOF OF THEOREM 1

**Theorem 1.** *Let $x \in \mathbb{R}^d$ with $x \neq 0$ and let $g : \mathbb{R}^d \to \mathbb{R}$ be the ReLU-network of the binary discriminator and denote by $\{Q_r\}_{r=1}^R$ the finite set of polytopes such that $g$ is affine on these*

*polytopes which exists by Lemma 1. Denote by $Q_t$ the polytope such that $\beta x \in Q_t$ for all $\beta \geq \alpha$ and let $x^{(L-1)}(z) = Uz + d$ with $U \in \mathbb{R}^{n_{L-1} \times d}$ and $d \in \mathbb{R}^{n_{L-1}}$ be the output of the pre-logit layer of $g$ for $z \in Q_t$. If $Ux \neq 0$, then*

$$\lim_{\beta \to \infty} \hat{p}(y|\beta x) = \frac{1}{K}.$$

*Proof.* We note that with a similar argument as in the derivation of (3) it holds

$$\hat{p}(y|\beta x) \leq \hat{p}(i|\beta x) + \frac{1}{K}\big(1 - \hat{p}(i|\beta x)\big) = \frac{K-1}{K}\hat{p}(i|\beta x) + \frac{1}{K}. \tag{12}$$

For all $\beta \geq \alpha$ it holds that $\beta x \in Q_t$ so that

$$\hat{p}(i|\beta x) = \frac{1}{1 + e^{-g(\beta x)}} = \frac{1}{1 + e^{\left\langle W_g^{(L_g)}, U\beta x + d\right\rangle + b_g^{(L_g)}}}.$$

As $x_i^{(L-1)}(x) \geq 0$ for all $x \in \mathbb{R}^d$ it has to hold $(\beta Ux + d)_i \geq 0$ for all $\beta \geq \alpha$ and $i = 1, \ldots, n_{L-1}$. This implies that $(Ux)_i \geq 0$ for all $i = 1, \ldots, n_{L-1}$ and since $Ux \neq 0$ there has to exist at least one component $i^*$ such that $(Ux)_{i^*} > 0$. Moreover, $W_g^{(L_g)}$ has strictly negative components and thus for all $\beta \geq \alpha$ it holds

$$g(\beta x) = \left\langle W_g^{(L_g)}, U\beta x + d\right\rangle + b_g^{(L_g)} = \beta\left\langle W_g^{(L_g)}, Ux\right\rangle + \left\langle W_g^{(L_g)}, d\right\rangle + b_g^{(L_g)}.$$

As $\left\langle W_g^{(L_g)}, Ux\right\rangle < 0$ we get $\lim_{\beta \to \infty} g(x) = -\infty$ and thus

$$\lim_{\beta \to \infty} \hat{p}(i|\beta x) = 0.$$

Plugging this into (12) yields the result. $\qquad\square$

## D EXPERIMENTAL DETAILS

**Datasets** We use CIFAR10 and CIFAR100 (Krizhevsky & Hinton, 2009) (MIT license), SVHN (Netzer et al., 2011) (free for non-commercial use), LSUN (Yu et al., 2015) (no license), the ILSVRC2012 split of ImageNet (Deng et al., 2009; Russakovsky et al., 2015) (free for non-commercial use), FGVC-Aircraft (Maji et al., 2013) (free for non-commercial use), Stanford Cars (Krause et al., 2013) (free for non-commercial use), OpenImages v4 (Kuznetsova et al., 2020) (images have a CC BY 2.0 license), Oxford 102 Flower (Nilsback & Zisserman, 2008) (no license) as well as 80 million tiny images (Torralba et al., 2008) (no license given, see also App. E). For the train/test splits we use the standard splits, except on 80M Tiny Images where we treat a random but fixed subset of 1000 images in the first 1,000,000 as our test set. For all datasets that get used as a test out-distribution we use a random but fixed subset of 1000 images.

Following (Bitterwolf et al., 2020), the smooth noise that is used is generated as follows. Uniform noise is generated and then smoothed using a Gaussian filter with a width that is drawn uniformly at random in $[1, 2.5]$. Each datapoint is then shifted and scaled linearly to ensure full range in $[0, 1]$, i.e. $x' = \frac{x - \min(x)}{\max(x) - \min(x)}$.

**Binary Training** The architecture that we use for the binary discriminator is relatively shallow (5 linear layers). The architecture is shown in Table 3. Our results are fairly robust to the exact choice of architecture and significantly larger models do not necessarily lead to better results as we show in App. I. Similarly to (Zhang et al., 2020; Bitterwolf et al., 2020), we use long training schedules, running Adam for 1000 epochs, with an initial learning rate of $1e-4$ that we decrease by a factor of 5 on epochs $500, 750$ and $850$ and with a batch size of 128 from the in-distribution and 128 from the out-distribution (for R.ImgNet: 50 epochs with drops at 25, 35, 45, batch sizes 32). In order to avoid large losses we also use a simple ramp up schedule for the $\epsilon$ used in IBP and we downweight the out-distribution loss during the initial phase of training by a scalar $\kappa$. Both $\epsilon$ and $\kappa$ are increased linearly from 0 to their final values (0.01 and 1, respectively) over the first 300 epochs (for R.ImgNet over the first 25 epochs). Compared to the training of (Bitterwolf et al., 2020)

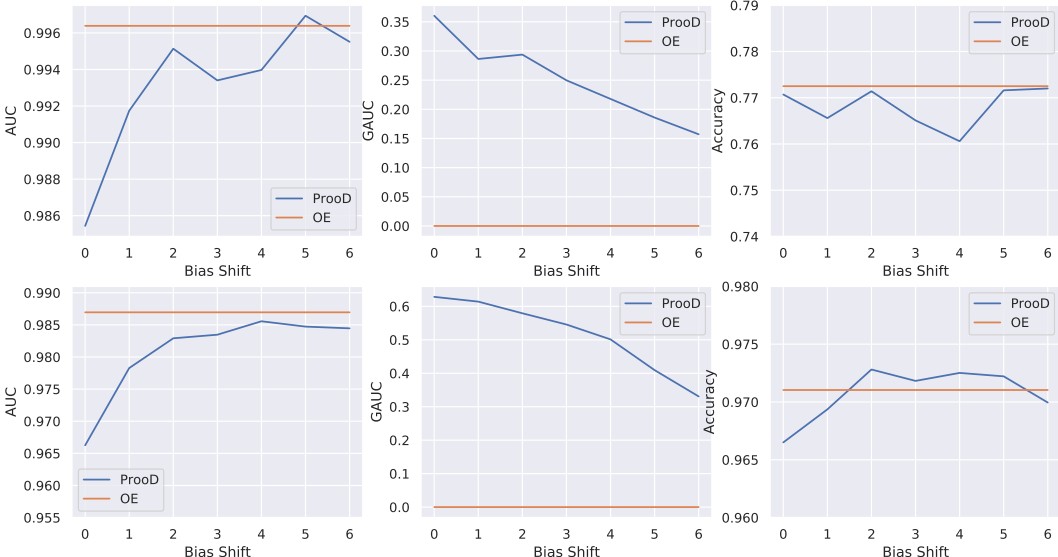

Figure 4: **Bias selection for CIFAR100 and RImgNet:** Using CIFAR100 (top) and R.ImgNet (bottom) as the in-distribution and the test set of OpenImages (or NotR.ImgNet respectively) as OOD we plot the test accuracy, AUC and GAUC as a function of the bias shift $\Delta$ (see Eq. (8)).

which sometimes fails, we found that training of the binary discriminator is very stable and even 100 epochs on CIFAR would be sufficient, but we found that longer training lead to slightly better results. Weight decay is set to $5 \cdot 10^{-4}$, but is disabled for the weights in the final layer. As data augmentation we use AutoAugment (Cubuk et al., 2019) for CIFAR and simple 4 pixel crops and reflections on R.ImgNet. The strict negativity of the weights leads to a negative bias of $g$ which can cause problems at an early stage if the $b_g^{(L_g)}$ is initialized at $0$ and thus we choose $3$ as initialization. All binary discriminators were trained on single 2080Ti GPUs, managed on a SLURM cluster. Overall, the training of a provable discriminator takes around 16h on CIFAR and 44h on R.ImgNet (wall clock time including evaluations and logging on each epoch).

**Semi-Joint Training** On CIFAR we train for 100 epochs using SGD with momentum of 0.9 and a learning rate of 0.1 that drops by a factor of 10 on epochs $50, 75$ and $90$ (on R.ImgNet 75 epochs with drops at 30 and 60). For all datasets we train using a batch size of 128 (plus 128 out-distribution samples in the case of OE). The CIFAR experiments were run on single 2080Ti GPUs. This takes about 4h20min in wall clock time. In order to fit batches of 128 in-distribution samples and 128 out-distribution samples on R.ImgNet we had to train using 4 V100 GPUs in parallel. Because of batch normalization in multi-GPU training it is important to not simply stack the batches but to interlace in- and out-distribution samples. The wall clock time was around 15h for the semi-joint training on R.ImgNet. For selecting the bias we use the procedure described in Section 4. The trade-off curves for the AUC and GAUC on CIFAR100 and R.ImgNet are given in Figure 4.

# E    80M TINY IMAGES AS TRAINING OUT-DISTRIBUTION

The 80M Tiny Images dataset has been retracted by the authors due to concerns over offensive class labels (Birhane & Prabhu, 2021). We support the decision of the community to move away from the use of 80M Tiny Images, so we choose to train our CIFAR models using a downscaled version of OpenImages v4 (Kuznetsova et al., 2020) as a training out-distribution. However, since all prior work used this dataset, we present results on 80M Tiny Images here in order to compare ProoD's performance to prior baselines. We encourage the community to use the results in Table 2 for future comparisons.

Our results are shown in Table 5. Apart from the models shown in Table 2 we add here the pre-trained energy-based OOD detector EB from (Liu et al., 2020) as an additional baseline for clean OOD

detection. As EB is not trained robustly, as expected EB has low AAUCs. In terms of clean OOD detection it performs similar to OE but with worse results on the more difficult OOD detection task CIFAR10 vs CIFAR100 and vice versa. For GOOD we use the pre-trained models from (Bitterwolf et al., 2020). For ATOM and ACET we use the pre-trained models from (Chen et al., 2020). Note that these use the densenet architecture and were actually trained to withstand attacks in the much stronger threat model of $\epsilon = \frac{8}{255}$. In the original paper, the authors claim near perfect AAUCs on this task but we can show that this is a case of gross overestimation of robustness. Even on the easier threat model $\epsilon = 0.01$ that we test in Table 5, their robustness is almost non-existent for both ATOM models and the CIFAR100 ACET model. The fact that their adversarial attacks were unable to find these samples clearly demonstrates that evaluating models adversarially is very difficult and potentially unreliable. Because of this we believe that our guarantees are a valuable contribution to the community.

Table 5: **Training with 80M Tiny Images:** We repeat the evaluation from Table 2 for models that were trained using 80M Tiny Images as out-distribution instead of OpenImages. Plain is identical to before and is just repeated for the reader's convenience. Note that the conclusions from the main paper still hold, which indicates that our method is robust to changes in the choice of training out-distribution. For ATOM and ACET we compare to pre-trained models from (Chen et al., 2020). Note that these models show almost no robustness on CIFAR100 - despite the far stronger claims in (Chen et al., 2020).

| In: CIFAR10 | Acc | CIFAR100 | | | SVHN | | | LSUN_CR | | | Smooth | | |
|---|---|---|---|---|---|---|---|---|---|---|---|---|---|
| | | AUC | GAUC | AAUC | AUC | GAUC | AAUC | AUC | GAUC | AAUC | AUC | GAUC | AAUC |
| Plain | 95.01 | 90.0 | 0.0 | 0.6 | 93.8 | 0.0 | 0.1 | 93.1 | 0.0 | 0.5 | 98.2 | 0.0 | 0.6 |
| OE | 95.53 | 96.1 | 0.0 | 6.0 | 99.2 | 0.0 | 0.4 | 99.5 | 0.0 | 15.2 | 99.0 | 0.0 | 11.3 |
| EB* | 95.22 | 93.8 | 0.0 | 2.8 | 99.3 | 0.0 | 0.0 | 99.5 | 0.0 | 6.0 | 99.4 | 0.0 | 3.5 |
| ATOM† | 95.20 | 93.7 | 0.0 | 14.4 | 99.6 | 0.0 | 8.6 | 99.7 | 0.0 | 40.0 | 99.6 | 0.0 | 18.8 |
| ACET† | 91.48 | 91.2 | 0.0 | 80.5 | 95.3 | 0.0 | 87.6 | 98.9 | 0.0 | 95.0 | 99.9 | 0.0 | 98.3 |
| GOOD$_{80}$* | 90.13 | 87.2 | 42.5 | 63.9 | 94.2 | 37.5 | 67.4 | 93.3 | 55.2 | 83.6 | 95.3 | 57.3 | 88.5 |
| GOOD$_{100}$* | 90.14 | 70.7 | 54.5 | 55.0 | 74.9 | 56.3 | 56.6 | 75.2 | 61.0 | 61.6 | 81.4 | 66.6 | 67.5 |
| ProoD-Disc | - | 67.4 | 61.0 | 61.7 | 73.2 | 65.5 | 66.4 | 78.0 | 72.2 | 72.7 | 82.3 | 71.5 | 72.9 |
| ProoD $\Delta=3$ | 95.47 | 96.0 | 41.9 | 43.9 | 99.5 | 48.8 | 49.4 | 99.6 | 47.6 | 53.1 | 99.7 | 55.8 | 57.0 |

| In: CIFAR100 | Acc | CIFAR10 | | | SVHN | | | LSUN_CR | | | Smooth | | |
|---|---|---|---|---|---|---|---|---|---|---|---|---|---|
| | | AUC | GAUC | AAUC | AUC | GAUC | AAUC | AUC | GAUC | AAUC | AUC | GAUC | AAUC |
| Plain | 77.38 | 77.7 | 0.0 | 0.3 | 81.9 | 0.0 | 0.2 | 76.4 | 0.0 | 0.3 | 88.8 | 0.0 | 0.5 |
| OE | 77.28 | 83.9 | 0.0 | 0.8 | 92.8 | 0.0 | 0.1 | 97.4 | 0.0 | 4.6 | 97.6 | 0.0 | 0.9 |
| EB* | 75.70 | 77.4 | 0.0 | 0.8 | 96.5 | 0.0 | 0.0 | 96.7 | 0.0 | 5.9 | 98.9 | 0.0 | 4.3 |
| ATOM† | 75.06 | 64.3 | 0.0 | 0.2 | 93.6 | 0.0 | 0.2 | 97.5 | 0.0 | 9.3 | 98.5 | 0.0 | 15.0 |
| ACET† | 74.43 | 79.8 | 0.0 | 0.2 | 90.2 | 0.0 | 0.0 | 96.0 | 0.0 | 2.1 | 92.9 | 0.0 | 0.3 |
| ProoD-Disc | - | 53.8 | 50.3 | 50.4 | 73.1 | 69.8 | 69.9 | 68.1 | 63.8 | 64.0 | 67.2 | 63.8 | 63.9 |
| ProoD $\Delta=1$ | 76.79 | 80.5 | 23.1 | 23.2 | 93.7 | 33.9 | 34.0 | 97.2 | 29.6 | 30.4 | 98.9 | 29.7 | 31.3 |

* Pre-trained WideResnet from (Liu et al., 2020).
†Densenet architecture, using models from (Chen et al., 2020) pre-trained with $\epsilon = \frac{8}{255}$.
*CNN architecture using pre-trained models from (Bitterwolf et al., 2020).

# F    FALSE POSITIVE RATES

Since in a practical setting a threshold for OOD detection ultimately has to be chosen, it can be interesting to study the false positive rate at a fixed threshold. It is relatively standard to pick the false positive rate at 95% true positive rate (called FPR in Table 6), where low values are desirable. We show the results for all methods and datasets in Table 6. Although ProoD has similarly good performance as OE on this task, it fails to give non-trivial guarantees. Therefore achieving stronger bounds on the worst-case FPR is an interesting task for future work.

Table 6: **False Positive Rates:** For all models we report accuracy on the test of the in-distribution and the false positive rate at 95% true positive rate (FPR) (smaller is better). We also show the adversarial FPR (AFPR) and the guaranteed FPR (GFPR) for different test out-distributions. The radius of the $l_\infty$-ball for the adversarial manipulations of the OOD data is $\epsilon = 0.01$ for all datasets. The bias shift $\Delta$ that was used for ProoD is shown for each in-distribution. ProoD struggles to give non-trivial guarantees for the FPR@95% on most datasets. However, different from GOOD or ProoD-Disc, the clean performance is generally as good as that of OE.

| In: CIFAR10 | | CIFAR100 | | | SVHN | | | LSUN_CR | | | Smooth | | |
|---|---|---|---|---|---|---|---|---|---|---|---|---|---|
| | Acc | FPR | GFPR | AFPR | FPR | GFPR | AFPR | FPR | GFPR | AFPR | FPR | GFPR | AFPR |
| Plain | 95.01 | 56.3 | 100.0 | 100.0 | 40.7 | 100.0 | 100.0 | 46.7 | 100.0 | 100.0 | 10.6 | 100.0 | 100.0 |
| OE | 94.91 | 52.2 | 100.0 | 99.9 | 15.4 | 100.0 | 100.0 | 0.0 | 100.0 | 99.0 | 0.0 | 100.0 | 99.0 |
| ATOM | 93.63 | 73.4 | 100.0 | 98.8 | 33.9 | 100.0 | 100.0 | 85.3 | 100.0 | 100.0 | 0.0 | 100.0 | 86.1 |
| ACET | 93.43 | 65.4 | 100.0 | 99.5 | 3.0 | 100.0 | 99.8 | 62.7 | 100.0 | 100.0 | 0.0 | 100.0 | 89.0 |
| GOOD$_{80}$ | 87.39 | 65.1 | 100.0 | 84.8 | 26.8 | 100.0 | 48.8 | 6.0 | 100.0 | 24.7 | 19.6 | 100.0 | 52.8 |
| GOOD$_{100}$ | 86.96 | 84.8 | 100.0 | 99.3 | 87.9 | 100.0 | 99.7 | 66.0 | 100.0 | 99.0 | 68.6 | 100.0 | 98.2 |
| ProoD-Disc | - | 83.9 | 87.5 | 87.2 | 76.9 | 84.9 | 84.2 | 76.7 | 85.3 | 84.7 | 96.6 | 98.4 | 98.4 |
| ProoD $\Delta{=}3$ | 94.99 | 48.0 | 99.9 | 99.7 | 9.1 | 100.0 | 100.0 | 0.0 | 100.0 | 98.7 | 0.0 | 100.0 | 100.0 |

| In: CIFAR100 | | CIFAR10 | | | SVHN | | | LSUN_CR | | | Smooth | | |
|---|---|---|---|---|---|---|---|---|---|---|---|---|---|
| | Acc | FPR | GFPR | AFPR | FPR | GFPR | AFPR | FPR | GFPR | AFPR | FPR | GFPR | AFPR |
| Plain | 77.38 | 80.1 | 100.0 | 100.0 | 77.3 | 100.0 | 100.0 | 79.0 | 100.0 | 100.0 | 70.0 | 100.0 | 100.0 |
| OE | 77.25 | 81.8 | 100.0 | 100.0 | 37.7 | 100.0 | 100.0 | 0.0 | 100.0 | 99.7 | 0.0 | 100.0 | 100.0 |
| ATOM | 68.32 | 81.3 | 100.0 | 99.6 | 51.0 | 100.0 | 97.4 | 26.3 | 100.0 | 94.7 | 10.0 | 100.0 | 86.0 |
| ACET | 73.02 | 87.9 | 100.0 | 100.0 | 8.2 | 100.0 | 100.0 | 87.0 | 100.0 | 100.0 | 0.0 | 100.0 | 98.2 |
| ProoD-Disc | - | 95.9 | 97.5 | 97.5 | 91.3 | 92.8 | 92.7 | 91.7 | 95.3 | 95.0 | 100.0 | 100.0 | 100.0 |
| ProoD $\Delta{=}5$ | 77.16 | 82.2 | 100.0 | 100.0 | 37.6 | 100.0 | 100.0 | 0.0 | 100.0 | 99.3 | 3.4 | 100.0 | 100.0 |

| In: R.ImgNet | | Flowers | | | FGVC | | | Cars | | | Smooth | | |
|---|---|---|---|---|---|---|---|---|---|---|---|---|---|
| | Acc | FPR | GFPR | AFPR | FPR | GFPR | AFPR | FPR | GFPR | AFPR | FPR | GFPR | AFPR |
| Plain | 96.34 | 55.2 | 100.0 | 100.0 | 48.2 | 100.0 | 100.0 | 75.2 | 100.0 | 100.0 | 0.0 | 100.0 | 100.0 |
| OE | 97.10 | 18.2 | 100.0 | 100.0 | 0.2 | 100.0 | 100.0 | 0.0 | 100.0 | 100.0 | 0.0 | 100.0 | 100.0 |
| ProoD-Disc | - | 59.2 | 65.2 | 65.0 | 51.0 | 67.8 | 66.5 | 51.7 | 63.7 | 62.3 | 100.0 | 100.0 | 100.0 |
| ProoD $\Delta{=}4$ | 97.25 | 18.5 | 100.0 | 100.0 | 0.5 | 100.0 | 100.0 | 0.0 | 100.0 | 100.0 | 0.0 | 100.0 | 100.0 |

# G   ADDITIONAL DATASETS

In addition to the results shown in Table 2, it is interesting to study how ProoD performs on additional datasets as well as the test set of the out-distribution it was trained on. For the CIFAR datasets, we report LSUN crops, LSUN_resize, Places365 (Zhou et al., 2017), iSUN (Xu et al., 2015), Textures (Cimpoi et al., 2014), 80M Tiny Images and uniform noise in Table 7. On RImgNet we report uniform noise and the training out-distribution in Table 8.

As in Table 2 the clean performance of ProoD is comparable to that of OE, but it achieves non-trivial GAUC. On CIFAR10, GOOD$_{100}$ achieves almost perfect GAUC against uniform noise, which comes at the price of significantly worse clean AUCs on all other out-distributions, see Table 2. Almost all methods achieve very high AAUCs on uniform noise, but it is not clear if a sufficiently powerful attack could lower those scores significantly. The unusually large gap between the GAUC and AAUC of ProoD would seem to indicate that that might be the case. Surprisingly, the robustness characteristics of both ATOM and ACET vary wildly between the different datasets, sometimes appearing to be perfectly robust and on other datasets displaying no robustness at all. Especially surprising are the relatively low AAUCs on the test set of the training out-distribution.

Table 7: **Additional Datasets:** We show the AUC, AAUC and GAUC for all models on uniform noise and on the test set of the train out-distribution.

| In: CIFAR10 | Acc | LSUN AUC | LSUN GAUC | LSUN AAUC | LSUN_resize AUC | LSUN_resize GAUC | LSUN_resize AAUC | Places365 AUC | Places365 GAUC | Places365 AAUC | iSUN AUC | iSUN GAUC | iSUN AAUC |
|---|---|---|---|---|---|---|---|---|---|---|---|---|---|
| Plain | 95.01 | 95.9 | 0.0 | 1.3 | 95.0 | 0.0 | 8.7 | 89.5 | 0.0 | 0.3 | 94.5 | 0.0 | 10.4 |
| OE | 94.91 | 98.7 | 0.0 | 0.9 | 97.4 | 0.0 | 6.1 | 99.9 | 0.0 | 2.5 | 97.6 | 0.0 | 9.4 |
| ATOM | 93.63 | 77.3 | 0.0 | 12.1 | 100.0 | 0.0 | 98.4 | 82.6 | 0.0 | 22.0 | 100.0 | 0.0 | 98.8 |
| ACET | 93.43 | 89.2 | 0.0 | 2.5 | 100.0 | 0.0 | 91.3 | 88.0 | 0.0 | 3.4 | 100.0 | 0.0 | 92.1 |
| GOOD80 | 87.39 | 96.5 | 68.4 | 89.7 | 87.4 | 61.7 | 69.5 | 96.8 | 58.8 | 90.2 | 87.1 | 58.9 | 70.3 |
| GOOD100 | 86.96 | 95.0 | 86.0 | 86.7 | 81.1 | 67.6 | 67.9 | 74.4 | 59.7 | 60.9 | 77.5 | 63.3 | 65.2 |
| ProoD-Disc | - | 95.8 | 94.1 | 94.2 | 76.4 | 70.3 | 71.5 | 76.6 | 71.1 | 71.5 | 74.9 | 69.0 | 70.3 |
| ProoD $\Delta=3$ | 94.99 | 99.2 | 82.2 | 82.3 | 97.1 | 57.7 | 59.2 | 99.9 | 58.7 | 59.6 | 97.1 | 56.4 | 58.1 |

| In: CIFAR10 | Acc | Uniform AUC | Uniform GAUC | Uniform AAUC | Textures AUC | Textures GAUC | Textures AAUC | 80M Tiny Images AUC | 80M Tiny Images GAUC | 80M Tiny Images AAUC | OpenImages AUC | OpenImages GAUC | OpenImages AAUC |
|---|---|---|---|---|---|---|---|---|---|---|---|---|---|
| Plain | 95.01 | 97.9 | 0.0 | 83.1 | 92.0 | 0.0 | 8.6 | 91.1 | 0.0 | 0.8 | 84.0 | 0.0 | 0.4 |
| OE | 94.91 | 99.9 | 0.0 | 98.2 | 99.9 | 0.0 | 13.8 | 93.6 | 0.0 | 0.6 | 99.7 | 0.0 | 2.5 |
| ATOM | 93.63 | 100.0 | 0.0 | 100.0 | 97.5 | 0.0 | 74.7 | 82.8 | 0.0 | 26.7 | 73.4 | 0.0 | 22.3 |
| ACET | 93.43 | 100.0 | 0.0 | 99.9 | 98.0 | 0.0 | 41.6 | 89.0 | 0.0 | 7.8 | 81.7 | 0.0 | 5.6 |
| GOOD80 | 87.39 | 99.1 | 83.9 | 98.1 | 96.1 | 54.7 | 89.9 | 84.7 | 50.5 | 66.8 | 93.6 | 53.7 | 85.6 |
| GOOD100 | 86.96 | 94.6 | 86.1 | 89.3 | 71.0 | 49.9 | 57.1 | 74.5 | 56.7 | 58.2 | 69.5 | 54.2 | 55.6 |
| ProoD-Disc | - | 53.8 | 46.7 | 46.7 | 75.3 | 69.9 | 70.5 | 75.7 | 69.8 | 70.9 | 64.8 | 59.0 | 59.6 |
| ProoD $\Delta=3$ | 94.99 | 99.9 | 35.0 | 94.7 | 99.9 | 57.6 | 63.1 | 93.8 | 57.5 | 59.0 | 99.8 | 47.7 | 49.7 |

| In: CIFAR100 | Acc | LSUN AUC | LSUN GAUC | LSUN AAUC | LSUN_resize AUC | LSUN_resize GAUC | LSUN_resize AAUC | Places365 AUC | Places365 GAUC | Places365 AAUC | iSUN AUC | iSUN GAUC | iSUN AAUC |
|---|---|---|---|---|---|---|---|---|---|---|---|---|---|
| Plain | 77.38 | 84.5 | 0.0 | 0.9 | 78.9 | 0.0 | 3.9 | 75.7 | 0.0 | 0.6 | 79.4 | 0.0 | 5.6 |
| OE | 77.25 | 94.5 | 0.0 | 1.3 | 88.2 | 0.0 | 2.1 | 99.7 | 0.0 | 1.9 | 88.3 | 0.0 | 3.1 |
| ATOM | 68.32 | 83.4 | 0.0 | 66.7 | 92.1 | 0.0 | 71.3 | 87.0 | 0.0 | 62.1 | 91.7 | 0.0 | 73.7 |
| ACET | 73.02 | 81.3 | 0.0 | 4.3 | 100.0 | 0.0 | 77.0 | 74.8 | 0.0 | 4.3 | 100.0 | 0.0 | 78.2 |
| ProoD-Disc | - | 81.4 | 78.8 | 79.3 | 70.1 | 66.7 | 67.1 | 71.7 | 68.2 | 68.5 | 69.3 | 65.6 | 66.0 |
| ProoD $\Delta=5$ | 77.16 | 94.8 | 28.5 | 28.7 | 86.3 | 22.6 | 22.8 | 99.7 | 23.6 | 23.8 | 87.2 | 22.1 | 22.4 |

| In: CIFAR100 | Acc | Uniform AUC | Uniform GAUC | Uniform AAUC | Textures AUC | Textures GAUC | Textures AAUC | 80M Tiny Images AUC | 80M Tiny Images GAUC | 80M Tiny Images AAUC | OpenImages AUC | OpenImages GAUC | OpenImages AAUC |
|---|---|---|---|---|---|---|---|---|---|---|---|---|---|
| Plain | 77.38 | 82.7 | 0.0 | 53.0 | 77.5 | 0.0 | 5.3 | 79.7 | 0.0 | 1.2 | 75.5 | 0.0 | 0.7 |
| OE | 77.25 | 99.3 | 0.0 | 91.4 | 99.0 | 0.0 | 7.5 | 80.2 | 0.0 | 1.5 | 99.6 | 0.0 | 1.0 |
| ATOM | 68.32 | 100.0 | 0.0 | 100.0 | 90.0 | 0.0 | 66.6 | 90.1 | 0.0 | 70.0 | 83.7 | 0.0 | 58.9 |
| ACET | 73.02 | 100.0 | 0.0 | 99.9 | 91.5 | 0.0 | 21.0 | 77.0 | 0.0 | 6.8 | 74.5 | 0.0 | 3.0 |
| ProoD-Disc | - | 40.9 | 37.0 | 37.4 | 53.6 | 50.2 | 48.1 | 59.7 | 56.7 | 56.9 | 58.3 | 54.6 | 54.8 |
| ProoD $\Delta=5$ | 77.16 | 99.7 | 12.1 | 87.0 | 99.1 | 16.8 | 22.6 | 80.5 | 19.5 | 19.7 | 99.7 | 18.6 | 20.1 |

Table 8: **Additional Datasets:** For RImgNet, we show the AUC, AAUC and GAUC for all models on uniform noise and on the test set of the train out-distribution, i.e. NotRImgNet.

| In: R.ImgNet | Acc | Uniform AUC | Uniform GAUC | Uniform AAUC | NotR.ImgNet AUC | NotR.ImgNet GAUC | NotR.ImgNet AAUC |
|---|---|---|---|---|---|---|---|
| Plain | 96.34 | 99.3 | 0.0 | 74.9 | 91.7 | 0.0 | 0.2 |
| OE | 97.10 | 99.6 | 0.0 | 84.6 | 98.7 | 0.0 | 1.2 |
| ProoD-Disc | | 99.7 | 99.2 | 99.3 | 73.6 | 69.9 | 69.9 |
| ProoD $\Delta=4$ | 97.25 | 99.8 | 79.7 | 95.2 | 98.6 | 50.1 | 51.3 |

## H    DERIVATION OF EQ. (3)

We restate the derivation in Eq. (3) in a slightly more verbose form. For all $y$ we have

$$\max_{\|x'-x\|_\infty \leq \epsilon} \hat{p}(y|x') = \max_{\|x'-x\|_\infty \leq \epsilon} \left( \hat{p}(y|x',i)\hat{p}(i|x') + \frac{1}{K}(1 - \hat{p}(i|x')) \right), \tag{13}$$

$$\leq \max_{\|x'-x\|_\infty \leq \epsilon} \left( \hat{p}(i|x') + \frac{1}{K}(1 - \hat{p}(i|x')) \right), \tag{14}$$

$$= \frac{K-1}{K} \max_{\|x'-x\|_\infty \leq \epsilon} \hat{p}(i|x') + \frac{1}{K}. \tag{15}$$

The inequality is obtained by using the trivial bound $\hat{p}(y|x,i) \leq 1$. This allows us to compute guarantees on the confidence of the entire model while treating the classifier itself entirely as a black-box. Because of this, there are no restrictions whatsoever on the architecture or training procedure that are used for fitting $\hat{p}(y|x,i)$.

## I    SIZE ABLATION FOR BINARY DISCRIMINATOR

Since larger models should typically lead to better performance, we investigated the impact that model size has on the performance of our binary discriminator. We retrained ProoD-Disc models with CIFAR10 as in-distribution and 80M Tiny images as the out-distribution (since our dataloader is faster than for OpenImages which speeds up training). Since longer schedules only slightly improve results we also used shorter schedules with only 300 epochs where $\epsilon$ and $\kappa$ linearly increase from 0 to 0.01 and 1.0 respectively within the first 100 epochs and the learning rate drops occur at 150, 200 and 250 epochs. As architectures we use 10 different CNNs with different widths and depths ranging from 5 to 8 layers (for the precise architectures please refer to sizes {S, XL_b, XS, SR, SR2, C1, C3s, C3, C2, C4} in the file `provable_classifiers.py` of the accompanying code). We present scatter plots of the models' their performance on CIFAR100 and 80M Tiny Images in Figure 5 against their size. Clearly, there is no correlation between model size and performance and most differences are rather small, justifying our choice of a fairly small architecture in the main paper.

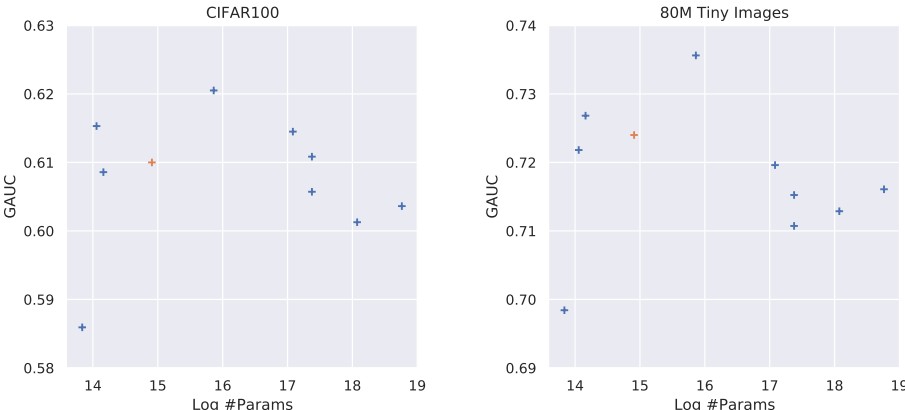

Figure 5: **Bigger models do not yield better guarantees:** We show scatter plots of the GAUC of different architectures against the log of the number of trainable parameters in the model. The orange cross indicates the architecture that is used in the main paper. There is no clear dependence of performance on model size, so it is preferable to use fairly small models.

## J    GENERALIZATION TO LARGER THREAT MODEL

Since $\epsilon = 0.01$ is a relatively weak threat model we evaluate if ProoD's guarantees actually generalize to the much stronger $\epsilon = \frac{8}{255} \approx 0.031$ that is standard in much of the literature on adversarial robustness. We use the exact same CIFAR models from Table 2 and show the results of our evaluation at $\epsilon = \frac{8}{255}$ in Table 9.

Table 9: **Generalization to larger** $\epsilon$**:** We evaluate all CIFAR models in Table 2 using an $\epsilon = \frac{8}{255}$, and thus an unseen threat model. The provable methods GOOD and ProoD generalize surprisingly well, while neither ATOM nor ACET display any generalization to the larger threat model.

| In: CIFAR10 | | CIFAR100 | | | SVHN | | | LSUN_CR | | | Smooth | | |
|---|---|---|---|---|---|---|---|---|---|---|---|---|---|
| | Acc | AUC | GAUC | AAUC | AUC | GAUC | AAUC | AUC | GAUC | AAUC | AUC | GAUC | AAUC |
| Plain | 95.01 | 90.0 | 0.0 | 0.0 | 93.8 | 0.0 | 0.0 | 93.1 | 0.0 | 0.0 | 98.0 | 0.0 | 0.0 |
| OE | 94.91 | 91.1 | 0.0 | 0.1 | 97.3 | 0.0 | 0.0 | 100.0 | 0.0 | 0.1 | 99.9 | 0.0 | 0.0 |
| ATOM | 93.63 | 78.3 | 0.0 | 1.3 | 94.4 | 0.0 | 1.5 | 79.8 | 0.0 | 0.2 | 99.5 | 0.0 | 9.6 |
| ACET | 93.43 | 86.0 | 0.0 | 1.1 | 99.3 | 0.0 | 1.1 | 89.2 | 0.0 | 0.8 | 99.9 | 0.0 | 3.8 |
| GOOD80* | 87.39 | 76.7 | 37.5 | 51.6 | 90.8 | 38.6 | 74.3 | 97.4 | 57.6 | 90.2 | 96.2 | 61.1 | 87.8 |
| GOOD100* | 86.96 | 67.8 | 39.4 | 43.5 | 62.6 | 29.0 | 30.9 | 84.9 | 67.6 | 70.7 | 87.0 | 63.3 | 69.2 |
| ProoD-Disc | - | 62.9 | 44.1 | 46.1 | 72.6 | 52.5 | 57.1 | 78.1 | 56.3 | 58.9 | 59.2 | 34.9 | 37.2 |
| ProoD $\Delta\!=\!3$ | 95.15 | 84.8 | 39.2 | 41.0 | 98.3 | 46.9 | 50.8 | 100.0 | 50.2 | 52.7 | 99.9 | 30.4 | 30.6 |

| In: CIFAR100 | | CIFAR10 | | | SVHN | | | LSUN_CR | | | Smooth | | |
|---|---|---|---|---|---|---|---|---|---|---|---|---|---|
| | Acc | AUC | GAUC | AAUC | AUC | GAUC | AAUC | AUC | GAUC | AAUC | AUC | GAUC | AAUC |
| Plain | 77.38 | 77.7 | 0.0 | 0.4 | 81.9 | 0.0 | 0.2 | 76.4 | 0.0 | 0.3 | 86.6 | 0.0 | 0.3 |
| OE | 77.25 | 77.4 | 0.0 | 0.2 | 92.3 | 0.0 | 0.0 | 100.0 | 0.0 | 0.7 | 99.5 | 0.0 | 0.5 |
| ATOM | 68.32 | 78.3 | 0.0 | 10.4 | 91.1 | 0.0 | 15.2 | 95.9 | 0.0 | 23.0 | 98.2 | 0.0 | 23.5 |
| ACET | 73.02 | 73.0 | 0.0 | 1.4 | 97.8 | 0.0 | 0.7 | 75.8 | 0.0 | 2.6 | 99.9 | 0.0 | 3.8 |
| ProoD-Disc | - | 56.1 | 41.1 | 43.1 | 61.0 | 50.5 | 51.8 | 70.4 | 57.5 | 58.8 | 29.6 | 20.9 | 20.8 |
| ProoD $\Delta\!=\!5$ | 76.51 | 76.6 | 13.7 | 14.1 | 91.5 | 16.9 | 16.9 | 100.0 | 18.1 | 18.2 | 98.9 | 8.1 | 8.1 |

| In: R.ImgNet | | Flowers | | | FGVC | | | Cars | | | Smooth | | |
|---|---|---|---|---|---|---|---|---|---|---|---|---|---|
| | Acc | AUC | GAUC | AAUC | AUC | GAUC | AAUC | AUC | GAUC | AAUC | AUC | GAUC | AAUC |
| Plain | 96.34 | 92.3 | 0.0 | 0.0 | 92.6 | 0.0 | 0.0 | 92.7 | 0.0 | 0.0 | 98.9 | 0.0 | 0.0 |
| OE | 97.10 | 96.9 | 0.0 | 0.2 | 99.7 | 0.0 | 0.0 | 99.9 | 0.0 | 0.0 | 98.0 | 0.0 | 0.0 |
| ProoD-Disc | - | 81.5 | 60.4 | 61.4 | 92.8 | 78.0 | 80.8 | 90.7 | 76.3 | 79.2 | 81.0 | 47.3 | 53.7 |
| ProoD $\Delta\!=\!4$ | 97.25 | 96.9 | 42.8 | 45.0 | 99.8 | 57.0 | 59.4 | 99.9 | 56.0 | 58.7 | 98.6 | 31.6 | 36.3 |

*Uses different architecture of classifier, see "Baselines" in Section 4.2.

Perhaps surprisingly, ProoD's guarantees generalize remarkably well to the much larger radius on CIFAR10. The same holds for GOOD, as was already observed in (Bitterwolf et al., 2020). On the other hand ATOM and ACET do not show any robustness at this radius, despite having lower accuracy, lower clean OOD performance and being more expensive to train. On the other hand ProoD's guarantees on CIFAR100 are quite weak at this radius, but given ATOM's and ACET's low AAUCs here and the fact that GOOD cannot be trained at all on CIFAR100, the results are not worse than for the competitors.

## K  ERROR BARS

In order to be mindful of our resource consumption we restrict the computation of error bars to our experiments on CIFAR10. Additionally, because the dataloader was much faster we ran these experiments using 80M Tiny Images as an out-distribution as opposed to OpenImages. We reran our experiments using the same hyperparameters 5 times. We computed the mean and the standard deviations for our models for all metrics shown in Table 5. The results are shown in Table 10. We see that the fluctuations across different runs are indeed rather small. Furthermore, the clean performance of OE and ProoD show no significant discrepancies.

Table 10: **Error Bars:** We show the mean and standard deviation $\sigma$ of all metrics for our CIFAR10 models across 5 runs. The tolerances for ProoD's clean performance are very small and yet the differences in clean performance between OE ProoD are not significant.

| In: CIFAR10 | | CIFAR100 | | | SVHN | | | LSUN_CR | | | Smooth | | |
|---|---|---|---|---|---|---|---|---|---|---|---|---|---|
| | Acc | AUC | GAUC | AAUC | AUC | GAUC | AAUC | AUC | GAUC | AAUC | AUC | GAUC | AAUC |
| Plain | 94.91 | 90.0 | 0.0 | 0.6 | 93.9 | 0.0 | 0.1 | 93.4 | 0.0 | 0.7 | 96.7 | 0.0 | 1.2 |
| Plain $\sigma$ | 0.16 | 0.1 | 0.0 | 0.1 | 1.2 | 0.0 | 0.0 | 0.3 | 0.0 | 0.2 | 2.1 | 0.0 | 0.5 |
| OE | 95.56 | 96.1 | 0.0 | 7.6 | 99.4 | 0.0 | 0.4 | 99.6 | 0.0 | 16.7 | 99.6 | 0.0 | 4.3 |
| OE $\sigma$ | 0.04 | 0.1 | 0.0 | 1.5 | 0.1 | 0.0 | 0.2 | 0.1 | 0.0 | 3.5 | 0.3 | 0.0 | 3.7 |
| ProoD-Disc | - | 67.7 | 61.6 | 62.2 | 75.5 | 68.6 | 69.3 | 76.5 | 70.4 | 70.9 | 87.2 | 77.7 | 78.8 |
| ProoD-Disc $\sigma$ | - | 0.7 | 0.7 | 0.7 | 1.4 | 1.7 | 1.5 | 1.4 | 1.7 | 1.7 | 3.6 | 4.3 | 4.3 |
| ProoD $\Delta=3$ | 95.60 | 96.0 | 42.2 | 44.1 | 99.4 | 48.6 | 49.2 | 99.6 | 47.1 | 52.0 | 99.8 | 55.2 | 57.0 |
| ProoD $\Delta=3$ $\sigma$ | 0.11 | 0.1 | 0.8 | 0.8 | 0.1 | 0.6 | 0.6 | 0.1 | 1.5 | 1.9 | 0.1 | 2.9 | 3.4 |

