# OpenReview forum: "Provably Robust Detection of Out-of-distribution Data (almost) for free"
_ICLR.cc/2022/Conference — ICLR 2022 Submitted_

### Official Review · Reviewer_6Xmu · 2021-10-24

**Correctness:** 3
**Technical Novelty And Significance:** 3
**Empirical Novelty And Significance:** 3
**Recommendation:** 8
**Confidence:** 5

**Main Review:**

By far, our main concern regarding the proposed solution is the following: Does the proposed method present (certified or not) adversarial robustness on the **in-distribution data**? For example, which part of the solution prevents a malicious person from using an attack on an **in-distribution example** (i.e., making the classifier believe that a sample belongs to the wrong class) to mislead the primary (i.e., the multiclass) classifier? The proposed binary classifier does not appear to be able to prevent this situation. The below excerpts from the paper make us suspect that the proposed solution does not present adversarial robustness on the **in-distribution data**.

1. _"In contrast to (Bitterwolf et al., 2020) this comes without loss in test accuracy or non-adversarial OOD detection performance as in our model the neural network used for the in-distribution classification task is independent of the binary discriminator. Thus, we p(yjx; i) have the advantage that the classifier can use arbitrary deep neural networks and is **not constrained to certifiable networks**."_

2. _"Note that this is not standard adversarial training for a binary classification problem as here we have an asymmetric situation: we want to be (certifiably) robust to adversarial manipulation on the out-distribution data **but not on the in-distribution** and thus the upper bound is only used for out-distribution samples."_

3. _"While in (Bitterwolf et al., 2020) they also used IBP to upper bound the confidence of the classifier this resulted in a bound that took into account all O(K2) logit differences between all classes. In contrast, our loss in Eq. (4) is significantly simpler as **we just have a binary classification problem and therefore only need a single bound**."_

Please, notice that ACET [2] presents adversarial robustness on the **in-distribution data**. Moreover, GOOD [3] presents _certified_ adversarial robustness on the **in-distribution data**. If we lose this, making the solution certified adversarial robust on the out-distribution data does not matter anymore, as the attacker may manipulate the solution by simply attacking in-distribution data rather than out-of-distribution data. Considering we are indeed correct, and the proposed solution does not provide certified adversarial robustness on the in-distribution data, **this fact may perfectly explain why the solution, unlike competing approaches, almost does not present loss in prediction accuracy**.

Maybe we are missing something. In such case, please, make the paper clearer regarding this point and show evidence that the proposed solution indeed presents (certified or not) adversarial robustness on the **in-distribution data**. For example, provide classification accuracy on adversarially manipulated in-distribution data (e.g., see [2]).

We have some additional less problematic concerns regarding the paper.

A drawback of the proposed method is the need to design an ad hoc binary classifier. We do not know whether the proposed binary classifier will work adequately for different models and datasets. Moreover, we need to define a training procedure for it. The solution adds a hyperparameter called the bias shift. We believe the authors could be more explicit about these limitations.

We also recommend that the authors combine their approaches with the IsoMax loss [4,5] and the IsoMax+ loss [6] rather than SoftMax loss or even OE to start with an improved OOD detection baseline. At least, the mentioned IsoMax loss could be cited as related works. The authors write: "we also want to achieve SOTA performance on unperturbed OOD data." However, IsoMax+ loss outperforms OE in some cases [6]. Hence, OE does not currently present SOTA performance.

The authors use the word "robust" in the paper to mean "adversarially robust." Considering that many other types of robustness exist, we suggest that authors write "adversarially robust" rather than simply "robust". Additionally, the authors sometimes refer to the multiclass classifier simply as the classifier. Considering that the solution also has a binary classifier, we suggest referring to the leading network as the multiclass classifier to make things more precise.

We recognize that the differences in performance of the proposed approach to the competing are significant. However, we need always to keep in mind that the proposed approach does not appear to present adversarial robustness on the **in-distribution data**, which is a major problem. Regardless of anything, it would be great to have the mean and standard deviation of five runs in Table 2. We also believe it is essential to add a column to Table 2 showing the classification accuracy on adversarial attacked in-distribution data.

Finally, we understand that the authors say that the approach is (almost) for free because it does not produce classification accuracy drop, and it has OOD detection performance similar to OE. However, many procedures need to be done to achieve this. Hence, we believe that "(almost) for free" may a bit be misleading.

[1] Deep Anomaly Detection with Outlier Exposure: https://arxiv.org/abs/1812.04606

[2] Why ReLU networks yield high-confidence predictions far away from the training data and how to mitigate the problem: https://arxiv.org/abs/1812.05720

[3] Certifiably Adversarially Robust Detection of Out-of-Distribution Data: https://arxiv.org/abs/2007.08473

[4] IsoMax loss: https://arxiv.org/abs/1908.05569

[5] IsoMax loss (journal): https://arxiv.org/abs/2006.04005

[6] IsoMax+ loss: https://arxiv.org/abs/2105.14399

David Macêdo

########################################################################################### ########################################################################################### ########################################################################################### ###########################################################################################

===== FINAL RECOMMENDATION POST-REBUTTAL ========

By considering that the proposed approach is somewhat novel and presents convincing results and that, after the rebuttal, the authors performed many runs for their method, recognized ACET is robust for in-distribution, added classification accuracy on attacked in-distribution data for all methods, improved clarity by adding terms proposed by the reviewer, make clear that the solution is not robust against attacks on in-distribution data, recognize that we need to design and train an ad-hoc binary discriminator; we are changing our recommendation for "accept".

########################################################################################### ########################################################################################### ########################################################################################### ###########################################################################################



**Summary Of The Paper:**

The paper proposes a method to provide certified adversarial robustness on the out-of-distribution. Moreover, almost no classification accuracy drop is observed. Furthermore, the detection performance on clean OOD is similar to Outlier Exposure (OE) [1] approach. Finally, the solution provably avoids the asymptotic overconfidence problem. The method is based on adding a binary classifier responsible for being certifiably robust to adversarial manipulation on the out-of-distribution data. Apparently, adversarial robust training is not applied to the multiclass classifier, preventing it from presenting classification accuracy drop. A semi-joint training is then applied.

**Summary Of The Review:**

Currently, it appears to us that the proposed solution does not provide adversarial robustness on the **in-distribution data**. If this is the case, we believe that this drawback makes the method much less valuable than previously published competing approaches. Suppose the authors clarify the text and prove that the proposed solution, like ACET and GOOD, provides adversarial robustness on the **in-distribution data**. In that case, we may improve our score mainly if the other concerns are also addressed.

---

> ### Author Response · Authors · 2021-11-21
> **Response to Reviewer 6Xmu**
>
> We want to thank the reviewer for their time and for offering constructive criticism on our paper.
>
> Addressing all concerns:
>
> "*Does the proposed method present (certified or not) adversarial robustness on the in-distribution data?*"\
> As the reviewer correctly points out, we openly state that our method is not meant to provide adversarial robustness around in-distribution samples. Rather, our method ensures that out-of-distribution samples that are correctly assigned a low confidence, will still only lead to low confidence predictions when adversarially manipulated. This means that our method will be able to flag out-of-distribution samples as OOD not only for typical samples, but even for worst-case examples. We believe that this is a crucial step towards trustworthy ML and that it is orthogonal to the desiderata of adversarially robust classification around the in-distribution. Note that in the detection setting, one does not necessarily want to adversarially robustly classify in-distribution samples with high confidence. The reason for this asymmetry is that for adversarially manipulated in-distribution samples, a clear statement cannot be made on whether they are in-distribution or out-of-distribution, whereas adversarially manipulated out-distributions are still from the out-distribution (also see our response to Reviewer fH7P).\
> However, it is very important to point out that neither ACET nor GOOD pursue or claim adversarially robust classification either.
> For GOOD, e.g. refer to the last sentence in their introduction: “In contrast to classifiers which have certified adversarial robustness on the in-distribution, GOOD [...]”
> Both are meant to address the exact same problem as our work.
>
> "*A drawback of the proposed method is the need to design an ad hoc binary classifier. We do not know whether the proposed binary classifier will work adequately for different models and datasets. Moreover, we need to define a training procedure for it. The solution adds a hyperparameter called the bias shift. We believe the authors could be more explicit about these limitations.*"\
> Training the binary discriminator is both simple and stable and we found a wide range of hyperparameters to work well. However, we agree that it does add some computational overhead to the model and also introduces a hyperparameter which we find negligible given the additional advantages of ProoD. Note that the hyperparameter is selected exclusively using measures computed from the training data. We have edited the writing to more explicitly point this out to the reader.
>
> "*We also recommend that the authors combine their approaches with the IsoMax loss [4,5] and the IsoMax+ loss [6] rather than SoftMax loss or even OE to start with an improved OOD detection baseline. At least, the mentioned IsoMax loss could be cited as related works.*"\
> At the time of submission we were unaware of the IsoMax loss. Since we mainly compare to methods that directly use the confidence score (Plain, OE, ACET, GOOD), we believe that combining the IsoMax loss with our method would be out-of-scope. However, our preliminary experiments confirm IsoMax's competitive performance on OOD detection and so we are happy to include it in our references
> It would indeed be interesting to combine this with the ideas of this paper but this would require a significantly different approach and is thus out of scope for this paper.

---

> > ### Author Response · Authors · 2021-11-21
> > **Response to Reviewer 6Xmu (cont.)**
> >
> > "*we suggest that authors write "adversarially robust" rather than simply 'robust'.*","*we suggest referring to the leading network as the multiclass classifier*"\
> > We appreciate the suggestions to make our paper more readable. We agree that different notions of robustness exist and, wherever possible, we have replaced “robust” with “adversarially robust” in the paper.
> >
> > In order to differentiate the multi-class classifier from the classifier between in- and out-distribution we refer to the binary model as a “binary discriminator” in the paper. We have improved the consistency of this terminology in the paper and now explicitly point out this distinction on page 3. If the reviewer believes that there is still room for confusion we are happy to make additional changes.
> >
> > "*it would be great to have the mean and standard deviation of five runs in Table 2.*"\
> > Unfortunately, since some of the methods are very expensive to train, we are unable to provide the standard deviation across several runs in this short time frame (due to the simultaneous CVPR deadline our GPU cluster was quite full). Note that ACET and ATOM employ adversarial training and GOOD requires 900 epochs on a relatively large network. Since our main focus lies on the guarantees without loss in classification accuracy, we believe that the results are convincing by themselves. However, for our own methods and the baselines Plain and OE, we did compute standard deviations across 5 runs on CIFAR10 using 80M TinyImages as an out-distribution. The results shown in Table 10 confirm that the fluctuations are quite small.
> > For the final version we will add mean and standard deviation using OpenImages instead.
> >
> > "*We also believe it is essential to add a column to Table 2 showing the classification accuracy on adversarial attacked in-distribution data.*"\
> > As we point out in the paper, adversarial robustness around the in-distribution is not a goal of either our method nor any of the listed competitors. We therefore don't list adversarially robust accuracy in Table 2. However, we have now updated the results section of our paper to explicitly reiterate that the methods do not pursue nor achieve adversarial robustness on the in-distribution.

---

> > ### Comment · Reviewer_6Xmu · 2021-11-24
> > **Brief comments**
> >
> > We appreciate the responses and are reevaluating our recommendation. We will probably improve our score.
> >
> > However, we still believe ACET, unlike GOOD and Proof, presents adversarial robustness for in-distribution (see table 1 in the paper), and this should be explicitly mentioned in the paper for more fair analyses and comparisons. A column for showing classification accuracy for in-distribution under attack would be good and fair.
> >
> > Despite the relevant contribution of the paper in protecting attacks against OOD data with convincing results, it has to become more explicit that we may still attack inliers.
> >
> > Hence, we think the contribution is significative, but we believe the paper may be a bit more clear in some points. Additionally, many executions of the same experiments would be great.

---

> > > ### Author Response · Authors · 2021-11-25
> > > **Brief response**
> > >
> > > Thanks for the quick response and we appreciate that the reviewer is considering our arguments.
> > >
> > > We will acknowledge ACET's in-distribution robustness in the final version and we will include the adversarial robust accuracy on the in-distribution for all methods.
> > >
> > > As requested we now show in the revised version the mean and standard deviation for our method’s performance across 5 executions in Table 10. The standard deviations are quite small. The table uses 80M Tiny Images as training out-distribution and we add mean and standard deviations for the runs on Open Images in the final version.

---

> ### Comment · Reviewer_6Xmu · 2021-11-26
> **===== FINAL RECOMMENDATION POST-REBUTTAL ========**
>
> We edit our review to include our final recommendation post-rebuttal.

---

### Official Review · Reviewer_guNu · 2021-10-24

**Correctness:** 2
**Technical Novelty And Significance:** 2
**Empirical Novelty And Significance:** 2
**Recommendation:** 3
**Confidence:** 4

**Main Review:**

I think this paper has the following strengths:

1. The authors derive the method ProoD in a principled way and prove that the method can prevent the asymptotic overconfidence of deep neural networks;

2.  They perform extensive experiments to evaluate the proposed method and also compare it to existing baselines;

3. The paper is well-written and easy to understand.

However, I think this paper has the following weaknesses:

1. The idea of training a discriminator independently via interval bound propagation (IBP) to achieve certified robustness on out-of-distribution samples is not very novel since IBP is an existing technique that can ensure certified robustness. Also, the theoretical result (Theorem 1) that the proposed joint classifier gets provably less confident in its decisions as one moves away from the training data is not entirely novel since it is mainly based on the previous theoretical results in the literature;

2. Since the proposed method ProoD uses IBP in training to get the certified discriminator g, it is expected that it has better certified robustness than other methods that don't use IBP. However, it might be hard to train models using IBP with a large perturbation budget (e.g. use $\epsilon=8/255$ that is common in previous OOD detection papers such as ATOM). The authors may need to acknowledge this limitation;

3. When the authors retrain ATOM and ACET models, they might need to consider using stronger PGD attacks during training since they use very strong attacks with adaptive step size for the evaluation of AAUC. In ATOM paper, they only use PGD attack with 5 steps and fixed step size. Using stronger PGD attacks with more steps and adaptive step size (e.g. use backtracking) for training ATOM and ACET might lead to better results (or AAUC) under their attacks in evaluation.

4. The performance of the proposed method ProoD is not stable across different in-distribution and OOD datasets. For example, in Table 2, the performance of ProoD on CIFAR-10 vs. Smooth is worse than that of ATOM and GOOD in terms of AAUC metric, and in terms of GAUC metric, the performance of ProoD is also worse than that of GOOD. The GAUC metric for ATOM and ACET seems meaningless since it is a lower bound and is equal to 0. On CIFAR-100 as the in-distribution dataset, it seems the performance of ProoD is usually worse than that of ATOM in terms of the AAUC metric. In Table 7 (in the appendix), on CIFAR-10 vs. Uniform or CIFAR-100 vs. Uniform, the performance of ProoD is also worse than that of existing methods like ATOM and GOOD. So it is unclear whether the performance of ProoD is better than that of existing methods or not. I think the authors should evaluate ProoD on more OOD datasets. In the ATOM paper, they also evaluate OOD detectors on OOD datasets like Textures, Places365, LSUN (resize), and iSUN. I suggest the authors report the performance of ProoD on these OOD datasets and compare it to existing methods.

5. I think the authors should give more details about attacking ProoD. For example, what's the attack objective they use to attack ProoD when evaluating the GAUC and AAUC? Since ProoD combines the classifier and detector, the adaptive attacks should attack both the classifier and the detector. If they only attack the detector when evaluating GAUC and AAUC, the results may not be correct and they need to re-evaluate them.



**Summary Of The Paper:**

In this paper, the authors propose ProoD which merges a certified binary classifier for in-versus out-distribution with a classifier for the in-distribution task in a principled fashion into a joint classifier. They show that ProoD simultaneously achieves three properties: 1) guaranteed OOD robustness via confidence upper bounds on $l_\infty$-balls around OOD samples; 2) it provably prevents the asymptotic overconfidence of deep neural networks; 3) it can be used with arbitrary architectures and has no loss in prediction performance and standard OOD detection performance. They perform extensive experiments to show the promising performance of the proposed method ProoD.

**Summary Of The Review:**

Although this paper proposes a principled method ProoD for robust OOD detection and has some interesting theoretical results, the experiments conducted are not enough to show the effectiveness of the proposed method. As I mentioned, the authors should be more careful in building the baselines and evaluating the methods. Also, they should give more details about the experiments like the attack objectives used. Thus, I think this paper is not ready for publication.


***[Post Rebuttal]***

I think the proposed method ProoD is not very novel since it simply combines previous techniques, and the performance of ProoD is not stable across different in-distribution and OOD datasets. Also, in practice, it might be hard to use ProoD since under the False Positive Rate at 95% true positive rate metric, ProoD doesn't have good performance (it might be hard to select a suitable threshold for ProoD). Thus, I keep my original score and think the paper is not ready for publication.

---

> ### Author Response · Authors · 2021-11-21
> **Response to Reviewer guNu**
>
> We want to thank the reviewer for their time and for offering constructive criticism on our paper.
>
> Addressing all concerns:
>
> "*1. The idea [...] is not very novel*", "*Also, the theoretical result (Theorem 1) [...] is not entirely novel*"\
> While our method does combine elements that were previously present in the literature, our empirical results show that combining them in this particular manner leads to a highly unexpected combination of properties, i.e. guarantees on adversarial robustness on the out-distribution without *any* loss in accuracy and almost no loss in clean OOD detection performance.
> Also note that while the derivation of Theorem 1 is proven using Lemma 1 of [Hein et. al 19], we prove for ProoD's exactly the opposite result. Instead of guaranteed asymptotic overconfidence as in [Hein et. al 19], we manage to guarantee asymptotically low confidence. In our opinion, the fact that achieving this is possible by using the specific properties of the binary discriminator we impose (strict negativity of the weights in the last layer) is thus novel and original. Thus we disagree with the reviewer and find this judgement overly negative.
>
> "*2. However, it might be hard to train models using IBP with a large perturbation budget (e.g. use that is common in previous OOD detection papers such as ATOM).*"\
> First of all, it is very important to point out that the cited work ATOM **did not** achieve any adversarial robust OOD detection for $\epsilon=8/255 \approx 0.031$ as we demonstrate in Table 5. Even for the strictly easier threat model of $\epsilon=0.01$ their AAUC are close to 0 in almost all cases. For example, their AAUC on CIFAR10 vs. SVHN is at most 8.6%, even though they claimed 99.6%. Therefore, the evaluation in their paper is flawed due to using too weak attacks at test time and thus the **adversarial robustness claims of their paper are unfortunately false**.
> We have already informed the first author of this paper about this and he agreed to update their tables (as soon as time permits) in order to more accurately reflect their method's performance.\
> Interestingly, the ProoD models that are trained on $\epsilon=0.01$ already generalize to exhibit non-trivial performance on $\epsilon=8/255$. Of course, these guarantees are weaker than on the easier threat model. We have added Appendix J where we show the evaluation for the threat model with $\epsilon=8/255$. In our experience, the training of ProoD is stable so that one can easily train models with a radius of $\epsilon=8/255$ without any loss in accuracy and without tweaking any hyperparameters (schedules etc.). We will include results for these models in the final version.
>
> "*3. When the authors retrain ATOM and ACET models, they might need to consider using stronger PGD attacks during training since they use very strong attacks with adaptive step size for the evaluation of AAUC. In the ATOM paper, they only use PGD attack with 5 steps and fixed step size. Using stronger PGD attacks with more steps and adaptive step size (e.g. use backtracking) for training ATOM and ACET might lead to better results*"\
> Using 5 steps of PGD was claimed by the authors of ATOM to be sufficient for adversarially robust OOD detection. We therefore just followed their paper. One can always argue that doing stronger attacks at training time can help but we feel that this discussion should have been part of the ATOM paper. Moreover, we would like to stress that neither ATOM or ACET aim to be provably adversarially robust but their main goal is improving empirical adversarial robustness of OOD detection.
> That being said, we retrained the ATOM  model (using their training protocol and code) using twice as many PGD-steps as recommended by them and found that the adversarial robustness on the out-distribution did indeed increase a bit as shown in the Table below.
>
> |    CIFAR10        |           | CIFAR100     |          | SVHN     |          | LSUN_CR     |          | Smooth     |          |
> |:-------------:    |-------    |:--------:    |:----:    |:----:    |:----:    |:-------:    |:----:    |:------:    |:----:    |
> |                   |  Acc      |    AUC       | AAUC     |  AUC     | AAUC     |   AUC       | AAUC     |   AUC      | AAUC     |
> |  ATOM 5 Steps     |  93.63     |     78.3     | 21.7     | 94.4     | 24.1     |    79.8     | 20.1     |   99.5     | 73.2     |
> | ATOM 10 Steps     | 93.52     |     82.5     | 35.8     | 98.9     | 86.4     |    78.3     | 25.3     |   99.9     | 97.4     |

---

> > ### Author Response · Authors · 2021-11-21
> > **Response to Reviewer guNu (cont.)**
> >
> > Note that training ATOM with the 10 step PGD attack takes twice as long to train as ProoD.
> > Nevertheless, ATOM has worse clean accuracy and worse clean AUCs compared to ProoD
> > and for the out-distributions CIFAR100 and LSUN_CR their empirical AAUCs are worse than our GAUCs. Given that adversarial attacks on OOD data are difficult as the models tend to have zero gradients and thus gradient-based optimization gets stuck (which lead to the overestimation of adversarial robustness in the ATOM paper), the reported numbers for the AAUC of ATOM could be still be overestimated, whereas our guarantees are provably correct.
> >
> > "*4. The GAUC metric for ATOM and ACET seems meaningless since it is a lower bound and is equal to 0.*"\
> > Unfortunately, ATOM and ACET are not certifiable and thus do not offer any non-trivial lower bounds. This is not surprising as this is also known for empirically adversarially robust models on the in-distribution where typically no provable guarantees are possible.  Thus our provable guarantees are a major advantage of ProoD over ATOM and ACET.
> >
> > "*So it is unclear whether the performance of ProoD is better than that of existing methods or not.*"\
> > Our goal is to have a method with the same clean accuracy and clean OOD detection performance as outlier exposure that additionally is provably adversarially robust against OOD detection. We think for practical acceptance of adversarially robust OOD detection methods it is key to maintain standard performance measures as well as possible.
> >
> > It is true that we do not outperform ATOM in terms of AAUC resp. GOOD in GAUC for all test out-distributions. But the point is that ATOM compared to ProoD has slightly worse clean accuracy for CIFAR10 but significantly worse clean OOD detection performance and for CIFAR100 it has similar clean OOD detection performance but yields a clean accuracy of 68.3% whereas we have 77.2% (please see Table 2). GOOD has most of the time better GAUCs than ProoD but has both for CIFAR10 and CIFAR100 significantly worse clean accuracy and OOD detection performance.
> >
> > "*I think the authors should evaluate ProoD on more OOD datasets. In the ATOM paper, they also evaluate OOD detectors on OOD datasets like Textures, Places365, LSUN (resize), and iSUN. I suggest the authors report the performance of ProoD on these OOD datasets and compare it to existing methods.*"\
> > We thank the reviewer for this suggestion and have added these additional datasets to the ones already reported in Appendix G. The additional results confirm the findings of Table 2. The clean OOD detection performance of ATOM for CIFAR10 is only better on LSUN_Resize, iSUN and Uniform with the largest gap being only 2.9% whereas ProoD is better on LSUN, Places365, Textures and 80M Tiny Images with the largest gap being 21.9%. Regarding AAUC the picture is mixed but as we stressed above this is not our main objective. For CIFAR100, the clean OOD detection results for the new datasets are mixed - the gaps are about 10% in terms of AUC in both directions. However, we highlight again, that the clean accuracy of ATOM on CIFAR100 is 8.84% worse than the one of ProoD.
> >
> > "*5. If they only attack the detector when evaluating GAUC and AAUC, the results may not be correct and they need to re-evaluate them.*"\
> > We agree with the reviewer in that attacks should always be adaptive to their models and so our attack jointly attacks both the detector and the classifier by directly maximizing the final confidence. This is described under ‘Guaranteed and Adversarial AUC’ in Section 4.2, where we now further clarified this attack objective. On top of that, please note that our GAUC provides provable lower bounds on the AAUC. As we point out in the paper, our bounds are so remarkably tight that it is **provably impossible** to come up with an attack that performs significantly better than our evaluation attack, when applied to ProoD. For example, no attack can possibly lower any of ProoD's AAUCs in Table 2 by more than 1%. On the other hand, the same cannot be said for the AAUCs of ATOM and ACET, for which there very well could be stronger attacks that lower the AAUCs even further.

---

> > ### Comment · Reviewer_guNu · 2021-11-24
> > **Major concerns**
> >
> > Thanks for the clarification! However, I still have some major concerns:
> >
> > 1. it seems the proposed method ProoD doesn't have clear advantages over the existing methods. The performance of ProoD is mixed: in some cases, its performance is better than the existing methods while in other cases, its performance is worse. Note that the baseline ATOM also has strong performance on the corrupted OOD inputs as shown in their paper. The way they use to generate the corrupted adversarial OOD examples is: for each OOD image, they generate 75 corrupted images and then select the one with the lowest OOD score (or highest confidence score to be in-distribution). So to construct corrupted adversarial OOD examples, we only need black-box attacks since we can enumerate all possible corruptions. I think the authors also need to evaluate their method ProoD under such corrupted OOD examples and compare it to ATOM. We want to know how ProoD will perform on various types of OOD inputs.
> >
> > 2. it is still not very clear to me how the authors compute GAUC and AAUC for ProoD. The authors clarify that their attack jointly attacks both the detector and the classifier by directly maximizing the final confidence. It would be better if the authors could write down the exact attack objective for attacking ProoD. Also, to compute GAUC for ProoD, it seems they also need to compute certified scores for $\hat{p}_f(y|x, i)$, where $f$ is the classifier network. Since the classifier is a complex neural network (e.g. ResNet), I am wondering how they compute the certified scores for such a complex network. If they could compute certified scores for the complex neural network, it seems they could also compute GAUC for ATOM and ACET. I think this point is very important and hope the authors could clarify it in detail.

---

> > > ### Author Response · Authors · 2021-11-25
> > > **Response to remaining concerns**
> > >
> > > Thanks a lot for the quick answer.
> > >
> > > “*it seems the proposed method ProoD doesn't have clear advantages over the existing methods*”\
> > > We reiterate the advantages of ProoD (please also refer to the second part of our original rebuttal; see below). It has no loss in test accuracy and (almost) no loss in clean OOD detection compared to Outlier exposure (OE), but additionally comes with certified guarantees for OOD detection under adversarial perturbations of the out-distribution inputs. None of the baselines simultaneously achieve this. OE is not adversarially robust, GOOD has significantly worse test accuracy and clean OOD detection, and **ATOM has no certified guarantees for adversarially robust OOD detection (clearly a major disadvantage since we could break their pretrained models) and is either worse in clean OOD detection (CIFAR10) or much worse in test accuracy (CIFAR100). We don’t understand how the reviewer thinks that there are no clear advantages of ProoD over ATOM and the other baselines.**
> > >
> > > “*Note that the baseline ATOM also has strong performance on the corrupted OOD inputs as shown in their paper*”\
> > > Note that the corruption robustness reported in the ATOM paper is not using an $l_\infty$-bounded threat model and, thus, is a different robustness task from the one pursued in our paper. However, we also benchmarked our CIFAR10 model using the same perturbations studied in the ATOM paper (on a subset of 1000 random but fixed samples from each out-distribution in Table 2). For a fair comparison, we compared to the retrained ATOM model because it uses the same training out-distribution and architecture as our ProoD and because the pre-trained model by the ATOM authors has *no* adversarial OOD robustness whatsoever.
> > >
> > > Our results for the models trained on Open Images are shown in the Table below. Our ProoD model out-performs ATOM by a large margin. We are happy to include these results in the final paper if the reviewers and the AC agree on this.
> > >
> > > |       |   CIFAR100 |   SVHN |   LSUN_CR |   Smooth |   Textures |   Places365 |   iSUN |   LSUN |   LSUN_resize |
> > > |:------|-----------:|-------:|----------:|---------:|-----------:|------------:|-------:|-------:|--------------:|
> > > | ATOM  |       35.0   |   40.2 |      29.2 |     79.1 |       80.6 |        38.7 |   74.6 |    8.6 |          68.0   |
> > > | ProoD |       66.3 |   94.1 |      89.1 |     97.7 |       93.1 |        88.1 |   85.5 |   95.3 |          85.4 |
> > >
> > > We also compare our ProoD model that was trained using 80M Tiny Images to the pre-trained ATOM model and compare in the table below. The performance of ProoD is similar to that of ATOM on this task despite this ATOM model having **no adversarial robustness on OOD**.
> > >
> > > |       |   CIFAR100 |   SVHN |   LSUN_CR |   Smooth |   Textures |   Places365 |   iSUN |   LSUN |   LSUN_resize |
> > > |:------|-----------:|-------:|----------:|---------:|-----------:|------------:|-------:|-------:|--------------:|
> > > | ATOM  |       76.7 |   91.3 |      95.5 |     98.6 |       96.4 |        92.1 |   94.4 |   94.9 |       93.9 |
> > > | ProoD |       83.2 |   94.3 |      93.0   |     96.6 |       96.0   |        91.1 |   92.0   |   95.0   |       91.7 |
> > >
> > >
> > >
> > >
> > > “*it is still not very clear to me how the authors compute GAUC and AAUC for ProoD*”\
> > > We gently remind the reviewer that we explain how to compute both quantities on page 7. For the reviewer’s convenience we reiterate it here. The AAUC and GAUC are computed as follows:
> > > $$AAUC_h(p_1, p_2) = E_{x \sim p_1, z \sim p_2}  (  1_{h(x)>h^{(lower)}(z)}   ) $$
> > > $$GAUC_h(p_1, p_2) = E_{x \sim p_1, z \sim p_2}  (  1_{h(x)>h^{(upper)}(z)}   ) $$
> > > Where $h^{(lower)}(z) \leq \max_{ |z-z'|\leq \epsilon } h(z') \leq h^{(upper)}(z)$ (names different from paper, because of markdown formatting limitations). $h^{(lower)}(z)$ can be computed by running an adversarial attack, like in the ATOM paper. Non-trivial bounds $h^{(upper)}(z)$ can only be computed for GOOD and ProoD (see answer below).
> > >
> > > “*It would be better if the authors could write down the exact attack objective for attacking ProoD*”\
> > > Please note that we explicitly state our attack objective on page 8: “For all attacks and all models we directly optimize the final score that is used for OOD detection.” For ProoD, this means that the attack objective is $\max_y \hat{p}(y|x).$
> > >
> > > “*it seems they also need to compute certified scores for $\hat{p}_f(y|x,i)$, where is the classifier network*”\
> > > No, as we use the trivial upper bound  $\max_y p(y|x,i) \leq 1$ in our upper bound on the confidence via Eq. (3)  (see Appendix H for more details). We therefore side-step the need to certify deep ResNets (for which no good certificates exist), which is one of the major innovations of our method. For ACET and ATOM no certificates are feasible similar to adversarially trained models on the in-distribution which are empirically robust but no certificates can be proven. However, for ProoD certificates are easy and fast to compute.

---

> > > > ### Comment · Reviewer_guNu · 2021-11-25
> > > > **Major concern**
> > > >
> > > > Thanks for the clarification! My major concern is still that the performance of ProoD is mixed. It doesn't always perform better than the baselines on different OOD test sets. Also, in practice, we need to pick a threshold for OOD detection. Thus, the False Positive Rate at 95% true positive rate (FPR) metric is also very important and I think the authors should report results under the FPR metric for **all evaluations**. I see that the authors report some results under the FPR metric in Appendix F. However, based on the results in Table 6 (Appendix F), it seems the performance of ProoD is not good and ProoD doesn't have clear advantages over the existing methods under the FPR metric. I think the FPR metric is more meaningful than the AUC metric and all previous works report main results under the FPR metric.

---

> > > > > ### Author Response · Authors · 2021-11-26
> > > > > **Response to remaining concern**
> > > > >
> > > > > It appears that the reviewer still refuses to acknowledge our paper’s central contribution. It is **not** to outperform every single method on every single metric. We also never claim this in the paper. Our goal is to provide guarantees on robustness to adversarial perturbations around OOD samples without any loss in clean accuracy and negligible loss in clean OOD detection performance. Our method shows that this is possible - a surprising result, as Reviewer U32o points out. All previous methods either do not provide guarantees or have severely diminished clean performance.
> > > > >
> > > > > That being said, we are more than happy to additionally report the FPR@95% for Table 5, 7, 8, 9 and 10 in the final version.

---

> ### Comment · Reviewer_guNu · 2021-11-29
> **Post Rebuttal**
>
> After reading the rebuttal, some of my concerns have been addressed. However, I still have some major concerns: 1. I still think that the idea is not very novel and the proposed method ProoD simply combines previous techniques in the literature; 2. The performance of ProoD is not stable across different in-distribution and OOD datasets. Out-of-distribution is any distribution that is different from the in-distribution. Thus, it is important to show the good performance of the method across various OOD test datasets. But it seems the proposed method doesn't achieve this. Also, in practice, we need to pick a threshold for OOD detection. So the False Positive Rate at 95% true positive rate (FPR) metric is very important. I think the FPR metric is more meaningful than the AUC metric and all previous works report main results under the FPR metric. The authors should consider using the FPR metric as the main metric and report results under the FPR metric for all evaluations. However, the authors only show limited results under the FPR metric in Appendix F, and based on the results in Table 6 (Appendix F), it seems the performance of ProoD is not good, and ProoD doesn't have clear advantages over the existing methods under the FPR metric.
>
> In summary, I think the proposed method ProoD is not very novel since it simply combines previous techniques, and the performance of ProoD is not stable across different in-distribution and OOD datasets. Also, in practice, it might be hard to use ProoD since under the False Positive Rate at 95% true positive rate metric, ProoD doesn't have good performance (it might be hard to select a suitable threshold for ProoD). Thus, I keep my original score and think the paper is not ready for publication.

---

> > ### Author Response · Authors · 2021-11-30
> > **Response to Post Rebuttal edit and comment**
> >
> > Since the reviewer is basing their criticism on objectively wrong statements, we correct each of these wrong claims in this answer.
> >
> >  “*The performance of ProoD is not stable across different in-distribution and OOD datasets*”\
> > ProoD is very stable across different in- and out-distributions compared to the available baselines:
> > * ATOM and ACET sometimes train to be robust (at reduced accuracy) and sometimes not at all. In fact, our paper trains the first ever ATOM models that actually show any adversarial OOD robustness. Therefore, we do not see how one could argue that ATOM or ACET have more stable performance than ProoD.
> > * GOOD training is quite unstable and we (or any other paper) could not even train it at all on CIFAR100, so clearly one cannot argue that it is more stable.
> > * Also note that no competitors even exist for high-resolution images like R.ImgNet (the methods using adversarial attacks for training are prohibitively expensive at that scale).
> >
> > The reviewer’s argument is simply objectively wrong when comparing our model to the baselines. This is without even mentioning that ProoD has higher accuracy and faster training than all three of those methods.
> >
> > “*Thus, it is important to show the good performance of the method across various OOD test datasets*”\
> > On CIFAR, we demonstrate our results on 11 different test out-distributions compared to 6 in the ATOM paper and in the GOOD paper.
> >
> > Also, since the reviewer seems to be under the impression that ATOM does show good and stable performance while ProoD doesn’t, we remind the reviewer that ProoD’s mean AAUC across those 11 test-distributions is 61.4% with a std of 15.3%, while ATOM’s is 52.0% with a std of 36.7%. Clearly, ProoD not only has higher performance (despite also having higher accuracy, tight robustness certificates and faster and more stable training) but even displays far lower variance across the different test-distributions.
> >
> > “*all previous works report main results under the FPR metric*”\
> > This is incorrect. GOOD did not report FPR values as their main metric and the reported AFPR values of the ATOM paper are all wrong as their evaluation of adversarial robustness is flawed (Table 5). Their correct AFPR@95 values are  at 100% everywhere.
> >
> > “*However, the authors only show limited results under the FPR metric in Appendix F, and based on the results in Table 6 (Appendix F), it seems the performance of ProoD is not good, and ProoD doesn't have clear advantages over the existing methods under the FPR metric*”\
> > The claim that ProoD has bad performance on the FPR@95 metric is false. The performance in Table 6 is on par with that of OE (on average even better) and certainly far better than that of, for example, ATOM.
> >
> > If the reviewer is referring to the AFPR@95 values instead, then we should again point out that no prior work has shown good performance on this metric.
> >
> > ---
> >
> > We would also like to point out that asking for the FPR values only after the update-phase is over and then using that as an argument for rejection is against good scientific practice (especially given that we even explicitly offered to include 5 additional FPR tables at the reviewer’s request). The reviewer brought up a lot of points and we could explain or show in which ways ProoD performs better than previous methods, in particular ATOM, and it was never acknowledged. Even our central contribution - the guarantees at no loss in accuracy - seems to never have been considered at all by the reviewer. The reviewer seems not to be interested in an objective scientific discussion but just seeks to constantly come up with new arguments against the paper and does not even care that we could refute all their previous arguments. We kindly ask all other reviewers and the AC to take this into consideration when making their final decision.

---

> > > ### Comment · Reviewer_guNu · 2021-12-01
> > > **Concerns**
> > >
> > > My comments are based on the empirical results provided by the authors. If we look at results in the paper (e.g. Table 2), the performance of the proposed method ProoD is quite mixed across in-distribution and OOD datasets (that is what I mean by "stable"). For example, in Table 2, on CIFAR-10 vs. Smooth, the performance of ProoD is worse than that of ATOM and GOOD in terms of AAUC metric, and in terms of GAUC metric, the performance of ProoD is also worse than that of GOOD; On CIFAR-10 vs. LSUN_CR, the performance of ProoD is also worse than that of GOOD in terms of GAUC metric. On CIFAR-100 as the in-distribution dataset, it seems the performance of ProoD is usually worse than that of ATOM in terms of the AAUC metric. In Table 7 (in the appendix), when evaluating the method on more OOD test datasets, the performance of ProoD is also mixed. I acknowledge that ProoD has some advantages as the authors pointed out. However, we should not simply ignore the disadvantages. The authors emphasized that ProoD has higher accuracy, but when ProoD achieves higher accuracy, its adversarial robustness is worse. So there is a trade-off here. For robust OOD detection, I think we should consider both clean accuracy and adversarial robustness. However, it seems ProoD doesn't achieve good performance when considering both clean accuracy and adversarial robustness (or achieve a good trade-off).
> > >
> > > For the False Positive Rate at 95% true positive rate (FPR) metric, initially, I think if the methods perform well under the AUC metrics, then they should also perform well under the FPR metric. However, based on the results in Appendix F, it seems this is not the case. Since the authors only report results under the FPR metric in Appendix, I didn't notice it when I reviewed the paper. But during the rebuttal period, I went through the appendix and found this issue. That is why I raise the concern after the update phase. And I think this is a valid concern.

---

### Official Review · Reviewer_fH7P · 2021-10-28

**Correctness:** 4
**Technical Novelty And Significance:** 3
**Empirical Novelty And Significance:** 4
**Recommendation:** 6
**Confidence:** 4

**Main Review:**

# Strengths

- The formulation of Prood is novel, sound, well-motivated, and reasonable.
- The theoretical contribution (Theorem 1) seems valid.
- The presented empirical results seem promising.


# Weaknesses

- There is a large room to improve in terms of the paper's clarity. There are multiple points in the paper that needs to be made clearer.
    - In Table 1, "High clean OOD" is supposed to be "High clean OOD detection performance".
    - In Eq. (3), $y$ should be clarified. Does it hold for all $y$'s? Also, the first inequality needs to be explained.
- The paper can be made more self-contained by including some background knowledge. For example, clear definitions of GAUC and AAUC should be provided.

# Questions

- For me, it is actually surprising that this approach works. In principle, a supervised classifier trained to discriminate inliers and outliers in a specific dataset will not necessarily be able to successfully detect unseen outliers. Probably this is why the clean AUC of ProoD-Disc in Table 2 is somewhat low. How can OOD detection AUC be improved even if an under-performing component is incorporated into the model?
- From Eq. (7) and the text nearby, $p(y|x,i)$ is not trained for robustness and the only component that is robustified is the binary classifier. (Please correct me if I'm wrong.) How can the whole model be robust if only a part of the model is robustified?
- Adversarial attack can also be performed on inliers so that it is misclassified (as in conventional attacks). How does ProoD respond to such attacks?

**Summary Of The Paper:**

A novel certifiable OOD detector is described in the paper. The proposed method, ProoD, merges a binary classifier and a multi-class classifier in a clever way to produce an OOD detector robust to adversarial perturbation. The binary classifier is trained to discriminate inliers and outliers, while the multi-class classifier is trained to predict class labels under outlier exposure. ProoD achieves good multi-class classification performance, good outlier detection performance, and robustness against adversarial perturbations on outliers.

**Summary Of The Review:**

I vote to accept this paper because its contributions are clear, novel, and significant.

---

> ### Author Response · Authors · 2021-11-21
> **Response to Reviewer fH7P**
>
> We want to thank the reviewer for their time and for offering constructive criticism on our paper.
>
> Addressing all concerns:
>
> “*In Table 1*, "High clean OOD" *is supposed to be* "High clean OOD detection performance" ”\
> Thanks, this has been fixed.
>
> “*In Eq. (3), should be clarified. Does it hold for all 's? Also, the first inequality needs to be explained*”\
> Yes, Eq. (3) holds for all $y$. We have edited the text to reflect this and also added Appendix H in order to elaborate on the derivation of Eq. (3)
>
> “*clear definitions of GAUC and AAUC should be provided*”\
> We have expanded the definition of the GAUC and the AAUC that was given in the first paragraph of page 8.
>
> "*In principle, a supervised classifier trained to discriminate inliers and outliers in a specific dataset will not necessarily be able to successfully detect unseen outliers*"\
> We agree that a-priori there is no reason why the approach has to generalize to unseen out-distributions. However, to some extent the same issue is true for outlier exposure or any other work that relies on a training out-distribution in some way. This prior work has shown that if the training out-distribution is diverse (basically a proxy of the distribution of natural images), then this generalization does indeed happen. The surprising observation that generalization to unseen out-distributions is possible even for provable models was already made in GOOD and has now been corroborated by our work.
>
> "*How can OOD detection AUC be improved even if an under-performing component is incorporated into the model?*"\
> Boosting or ensemble techniques in general show that it is possible to combine weak classifiers to build a stronger classifier. For our OOD detection task,  one can deduce why this works from
> their combination (see Equation 2, where in our approach we fix $\hat{p}(y|x,o)=\frac{1}{K}$ and note that $\hat{p}(o|x)=1-\hat{p}(i|x)$):
>        $$ \hat{p}(y|x) = \hat{p}(y|x,i) \hat{p}(i|x) + \frac{1}{K} ( 1 - \hat{p}(i|x)).$$
> a) Clean OOD:  here the  classifier $\hat{p}(y|x,i)$ (trained similar to Outlier exposure) enforces already low confidence close to uniform on out-of-distribution points and thus irrespective of what kind of values $\hat{p}(i|x)$ has, the resulting output $\hat{p}(y|x)$ will be close to uniform as well and this explains why clean OOD detection works similar to Outlier exposure.
>
> b) Adversarial OOD: here the classifier $\hat{p}(y|x,i)$ part is potentially completely corrupted and close to 1 but now the binary discriminator kicks in and ensures that at least for a significant fraction of points $\hat{p}(i|x)$ is still low enough so that the resulting prediction $\hat{p}(y|x)$ is close enough to a uniform distribution  so that they can be distinguished from the in-distribution samples via the confidence.
> This is explains why the combination works significantly better than each individual part of it
> and leveraging this kind of redundancy in an effective way is the key advancement of ProoD.  Due to the page limit, incorporating this discussion into the main paper would require significantly shortening or omitting other parts of the paper, but if the Reviewer requests it we would be happy to do so for the final version.

---

> > ### Author Response · Authors · 2021-11-21
> > **Response to Reviewer fH7P (cont.)**
> >
> > "*How can the whole model be robust if only a part of the model is robustified?*", "*In Eq. (3), $y$ should be clarified. Does it hold for all $y$'s? Also, the first inequality needs to be explained.*"\
> > Thanks for the question, the upper bound holds for all classes $y$ (we have added this in text).
> > We have added a more detailed derivation of this bound in Appendix H.
> > The main point is as discussed in the answer above that the classifier plays no role here - actually we upper bound in the equation for $\hat{p}(y|x)$ (please see answer above) the term $\hat{p}(y|x,i)$ with 1, the worst case. Our guaranteed OOD detection is thus solely based on the performance of the binary discriminator which has been trained to be certifiably adversarially robust.
> >
> > "*Adversarial attack can also be performed on inliers so that it is misclassified (as in conventional attacks). How does ProoD respond to such attacks?*"\
> > As we point out on page 4, our goal is *not* to be robust to adversarial perturbations around inliers. The reason for this asymmetry is that for adversarially manipulated in-distribution samples, a clear statement cannot be made on whether they are in-distribution or out-of-distribution, whereas adversarially manipulated out-distributions are still from the out-distribution. Therefore, ProoD is indeed vulnerable to adversarial attacks on the inliers. As reviewer U32o correctly points out, if we wanted to achieve adversarial robustness around inliers, prior literature indicates that one could almost certainly not be achieved it without any loss in accuracy as then one needs either an adversarially robust classifier or the binary discriminator would need need to detect adversarial examples. However, the latter task has been shown by Tramer et al (see also answer to RU32o) to be as difficult as building an adversarially robust classifier.

---

### Official Review · Reviewer_U32o · 2021-11-02

**Correctness:** 2
**Technical Novelty And Significance:** 2
**Empirical Novelty And Significance:** 2
**Recommendation:** 6
**Confidence:** 4

**Main Review:**

**Strength**\
The proposed method is sensible and shows consistent improvements.\
The paper provides experiments on several detection benchmarks.

**Weakness**\
*Limited evaluation*\
Despite the fact that this scheme utilizes a certified classifier, the author evaluates with empirical robustness measures (to calculate the maximum perturbation). Rather than that, I suggest utilizing a certified robustness measure with $l_\infty$ extension (Zhang et al., 2021).
Additionally, considering AutoAttack (Croce et al., 2020) to attack the confidence $\max p(y|x)$ will be more convincing (as an empirical robustness measure): calculate the maximum confidence over the ensemble attacks in AutoAttack.

*Discussion with Tramer et al., 2021*\
Tramer et al., 2021 prove that detecting adversarial samples is hard as classification (Tramer et al., 2021), i.e., classifying $\epsilon$ is almost the same as detecting $2*\epsilon$ sample. Due to this paper, it was hard for me to believe that adversarial detection is (almost) free: as it is known to sacrifice the clean accuracy to obtain adversarial robustness (Zhang et al., 2019). Can the author rigorously discuss with the following paper (Tramer et al., 2021)?

*Limited technical novelty*
* The proposed method can be seen as a combination of two OOD methods (Hsu et al., 2020) and (Bitterwolf et al., 2020).
* The main technical novelty of this paper is to (1) model predictive distribution with in-and-out conditional distribution (2) utilize IBP for modeling in-and-out distribution to model certification of OOD robustness.
* However, I believe (1) can be found in (Hsu et al., 2020), and (2) corresponds to (Bitterwolf et al., 2020).

*The AUC of ProoD-disc seems to be low*. This implies that the discriminator does not capture the in-and-out distribution probability well.

(minor) *The writing and presentation can be improved*
* For instance, a single paragraph contains many messages, making the readers confused about the main message.

**Questions**\
Is there a reason for utilizing a small network for certified robustness? The robustness tends to increase by the network size (Xie et al., 2020).\
Is it possible to report the GOOD (Bitterwolf et al., 2020) result in CIFAR-100 and R.ImgNet? I believe it is the main baseline to consider.

**References**\
Zhang et al., 2019 “Theoretically Principled Trade-off between Robustness and Accuracy”\
Bitterwolf et al., 2020, “Certifiably adversarially robust detection of out-of-distribution data”\
Croce et al., 2020, “Reliable evaluation of adversarial robustness with an ensemble of diverse parameter-free attacks”\
Hsu et al., 2020, “Generalized ODIN: Detecting Out-of-distribution Image without Learning from Out-of-distribution Data”\
Xie et al., 2020, “Intriguing properties of adversarial training at scale”\
Tramer et al., 2021, “Detecting Adversarial Examples Is (Nearly) As Hard As Classifying Them”\
Zhang et al., 2021, “Towards Certifying L-infinity Robustness using Neural Networks with L-inf-dist Neurons”

**Summary Of The Paper:**

This paper aims to detect out-of-distribution data in an adversarially robust manner. To this end, the authors incorporate a certified (binary) classifier to model in-and-out distribution and jointly model predictive distribution $p(y|x)$ by conditioning with the binary classifier, i.e., $p(y|x, in)p(in|x) + p(y|x,out)p(out|x)$. The authors show that the proposed method is empirically strong under various detection scenarios.

**Summary Of The Review:**

I recommend weak rejection for this review. I believe the evaluation is somewhat questionable and also needs some rigorous discussion with related works. I still believe the idea is sensible, so I carefully request the authors to respond to my weakness part during the rebuttal.

-------
**POST REBUTTAL:** After the response, I am slightly above the threshold as the proposed method seems to be a scalable work as it stabilizes the hardness of training robust out-of-distribution (OOD) detector.

---

> ### Author Response · Authors · 2021-11-21
> **Response to Reviewer U32o**
>
> We want to thank the reviewer for their time and for offering constructive criticism on our paper.
>
> Addressing all concerns:
>
> "*Rather than that, I suggest utilizing a certified robustness measure with extension (Zhang et al., 2021).*"\
> We agree with the reviewer that empirical robustness evaluation alone is not sufficient for judging adversarial robustness claims. However, we do, in fact, use a certified robustness measure (like Zhang et al., 2021) by computing the GAUC (see the first paragraph on page 8), which provides a provable lower bound on the worst-case AUC within the $l_\infty$-threat model (which is achieved by applying IBP to compute the confidence score, see Section 4.2).
> Note also that the gap between the AAUC (which is an upper bound on the GAUC, computed by strong adversarial attacks, see Section 4.2) and the GAUC is very small, showing that none of them can be significantly improved anymore.
>
> "*Additionally, considering AutoAttack (Croce et al., 2020) to attack the confidence will be more convincing*"\
> We agree that AutoAttack is the de-facto standard for evaluating adversarial robustness, and our evaluation uses the applicable attacks of the AutoAttack ensemble adapted to the evaluated methods. Such adaptations are necessary since AutoAttack was created for the evaluation of adversarial robustness on the in-distribution (which we do not pursue in this paper, see below) and is therefore not directly applicable to our problem as we need to maximize as objective the confidence. Additionally, note that we do already employ AutoPGD on our objective in addition to several restarts of monotone PGD with adaptive step-size selection and backtracking. Out of the other attacks in AutoAttack, FAB attack can't directly be adapted to our setting as it searches for the decision boundary which is not of interest to our setting, and SquareAttack is a black-box attack and therefore generally performs much worse than PGD-based attacks. Nonetheless, for the updated draft we have added an adapted version of SquareAttack to our evaluation pipeline, although this did not change any of the results.
>
> "*Discussion with Tramer et al., 2021 [and (Zhang et al., 2019)]*"\
> Note that both these works focus on adversarial examples on the in-distribution. It is a well-accepted fact that achieving adversarial robustness on the in-distribution generally comes at some cost in clean accuracy and Tramer et al. suggests that we should expect a similar phenomenon to hold for models that adversarially robustly classify with a reject-option. Our work is different in that we do not claim any adversarial robustness on the in-distribution, and in particular we do not detect adversarial samples (in the classical sense of perturbing in-distribution samples to change the class decision). Instead we only reject samples from an out-distribution, even if these have been adversarially modified. We agree with the reviewer's sentiment that our result is quite surprising in light of previous work and we believe this to be our greatest contribution: showing that in the case of OOD data, adversarial robustness and even provable adversarial robustness can be achieved at no cost in clean accuracy and only slight cost in clean OOD detection performance.

---

> > ### Author Response · Authors · 2021-11-21
> > **Response to Reviewer U32o (cont.)**
> >
> > "*Limited technical novelty*"\
> > Even though elements of our method were already present in some prior works, we clearly demonstrate that our specific way of combining them leads to novel and (as you pointed out in the previous question) surprisingly strong results. No other method simultaneously achieves provably adversarially robust OOD detection without any loss in accuracy and no other method simultaneously achieves adversarially robust OOD detection and provably low confidence asymptotically. One important step that is novel in our paper is the splitting up of the provable adversarial robustness part and the classification part, which allows the latter to maintain its full expressive power and therefore good performance. To us, the fact that our method is straightforward to implement is a feature and we believe it should not be held against the paper.
> >
> > "*The AUC of ProoD-disc seems to be low. This implies that the discriminator does not capture the in-and-out distribution probability well.*"\
> > We agree that the clean AUC of ProoD-Disc is fairly low. This is actually partially due to the IBP training. For example, training the same architecture using an $\epsilon=0$ (i.e. clean training) leads to an AUC of 74.0% on CIFAR10 vs CIFAR100 (instead of 62.9%) and 90.0% on CIFAR10 vs OpenImages (instead of 58.3). Of course this model has no guarantees on adversarial robustness. This seems analogous to the conventional wisdom that adversarial robustness lowers clean performance. Only when combining the binary discriminator with the classifier using our proposed method do we achieve the best of both worlds: non-trivial guarantees at almost no loss in clean OOD detection performance.
> >
> > "*Is there a reason for utilizing a small network for certified robustness? The robustness tends to increase by the network size (Xie et al., 2020).*"\
> > We did run ablations for the model size of ProoD-Disc and found that the results did not improve with model size. We have added this ablation as an additional appendix.
> >
> > "*Is it possible to report the GOOD (Bitterwolf et al., 2020) result in CIFAR-100 and R.ImgNet? I believe it is the main baseline to consider.*"\
> > As we discuss in the "Baselines'' section of our paper, training GOOD is quite unstable and we were unable to train their method on CIFAR100 (beyond 22% clean accuracy) and R.ImgNet (GOOD requires extensive tuning to find feasible schedules). We view the stability of our training as another strong argument in favor of ProoD over GOOD.

---

> > > ### Comment · Reviewer_U32o · 2021-11-26
> > > **Rebuttal response**
> > >
> > > I thank the author for their response, and it partially resolved my concerns.\
> > > However, there are still some parts that I want to discuss.
> > >
> > > *Can the author provide additional (ensemble of ) targeted attacks for the evaluation?* It will be truly meaningful.
> > > - I do understand that all attacks in AutoAttack (e.g., FAB) may not be applicable for this case. However, I am still questioning whether the targeted attacks will work, i.e., maximizing each class probability. Currently, it seems like the author only has considered maximizing the classifier's confidence, i.e., the maximum prediction probability. In the context of adversarial robustness, many defense schemes fail on the ensemble of targeted attacks (Croce et al., 2020).
> > >
> > > *Why the in-distribution adversarial example and out-of-distribution adversarial example should differ?*
> > > - It seems both are adversarial examples and out-of-distribution samples.
> > >
> > > *Is there any intuition (1) why ProoD is stable and (2) why GOOD is not?*
> > >
> > > *(minor) Some clarification on the evaluation*
> > > - The author mentioned that “For all attacks and all models we directly optimize the final score that is used for OOD detection” on page 8. What is the final score in this case..? I believe it might be equation (2) the $\hat{p}(y|x)$…?
> > >
> > > Croce et al., 2020, “Reliable evaluation of adversarial robustness with an ensemble of diverse parameter-free attacks.”

---

> > > > ### Author Response · Authors · 2021-11-26
> > > > **Re: Rebuttal response**
> > > >
> > > > We thank the reviewer for the response and appreciate the interesting questions.
> > > >
> > > >
> > > > *“Can the author provide additional (ensemble of ) targeted attacks for the evaluation?”*\
> > > > We agree that running targeted attacks against each class can be important for the evaluation of adversarial robustness. However, note that our paper focuses on provable adversarial robustness. Specifically, the GAUC is a guaranteed lower bound for the AAUC. The gap between AAUC and GAUC is very small for ProoD in Table 2. **Thus, there provably exists no attack that further degrades the AAUCs of our method in Table 2 by more than 1%**. The same cannot be said for any of the competing methods though, so their AAUCs may drop significantly. Since these additional attacks are computationally very expensive, we are unable to provide them in time for this response, but we would be happy to do so for the final version. However, note that the results could not possibly affect the central claim of our paper.
> > > >
> > > >
> > > > *“Why the in-distribution adversarial example and out-of-distribution adversarial example should differ?”*\
> > > > This is a very interesting question and we do not currently know why this is the case. One possible explanation is the following: if one wants to detect adversarial samples around the in-distribution as OOD, then at least some robustness in the classification is still required, because there is a gradual transition from ID to OOD, and some perturbations of ID samples are clearly still ID, e.g. very small perturbations. However, this does not hold for perturbations around OOD samples because every single point in the ball is OOD as well.
> > > >
> > > >
> > > > *“Is there any intuition (1) why ProoD is stable and (2) why GOOD is not?”*\
> > > > The GOOD objective simultaneously solves the classification task and the provably robust OOD detection task. The former increases the confidences while the latter decreases them. This means that both objectives need to be carefully balanced all throughout training or else the classifier “collapses” and becomes completely uniform. For ProoD, these conflicting goals are disentangled as the binary discriminator is responsible for adversarial OOD detection and the classifier is responsible for classification and clean OOD detection (see also answer to Reviewer fH7P).
> > > >
> > > >
> > > >
> > > >
> > > >
> > > > *“What is the final score in this case..? I believe it might be equation (2) the $\hat{p}(y|x)$”*\
> > > > This is correct. We will make the attack objective more explicit in the final version.

---

> > > > > ### Comment · Reviewer_U32o · 2021-11-29
> > > > > **RE: Re: Rebuttal response**
> > > > >
> > > > > Thank you for the response, and I am slightly above the acceptance.
> > > > >
> > > > > I still believe the performance of ProoD is limited which is also stated by Reviewer guNu and fH7P. However, I raised my point since I believe that the proposed method is scalable as it stabilizes the hardness of training robust out-of-distribution (OOD) detector.
> > > > >
> > > > > In the final revision, please add the discussions and evaluations that were in the rebuttal.

---

### Official Review · Reviewer_urW9 · 2021-11-03

**Correctness:** 3
**Technical Novelty And Significance:** 2
**Empirical Novelty And Significance:** 2
**Recommendation:** 5
**Confidence:** 4

**Main Review:**

Strengths:

1. Combining OOD detection and classifiers is an interesting idea and the proposed "OOD-aware" training can be effective to improve the robustness of the classifiers.

2. Authors provide a confidence bound for the classifier which is useful to develop certifiable robust OOD-aware models.

3. Experiments on the benchmark datasets show that the proposed approach achieves comparatively high accuracy while maintaining a guarantee on the AUC score.


Weaknesses:

1. The draft lacks clarity in many aspects.

a. In table 1, metrics are not clear until section 3.

b. Figure 1 cannot be interpreted from the caption. It is not clear how the equations in the figure map to equations in the draft.

c. In the experiments section, "semi-joint training" requires more details. For example, how are the several models trained with binary shifts? Metrics in figure 2 are not clear from the associated text.

2.  Authors claim to achieve better guarantees than the existing approaches (e.g., CCU, GOOD). However, it is not clear why Theorem 1 entails a tighter bound than Lemma 1 (Hein 2019).

3. The bounds in Eq 5 appear trivial from the computations of W_{+}, and W_{-}. How do these bounds affect the tightness of the bounds in Eq 6.

4. I would expect the joint training would improve the performance of both OOD detection and the classifier. However, as shown in table 2, outlier exposure, which is not the best state of the art, still performs better. What are the authors' comments on this.

5. Authors are encouraged to compare with the more recent OOD detection approaches

a. Liu et al., "Energy-based out-of-distribution detection"

b. Lee et al., "Training confidence-calibrated classifiers for detecting out-of-distribution samples."

**Summary Of The Paper:**

This paper presents an approach to combine OOD detection and classifier to develop a robust OOD detection framework with high classification accuracy. The authors provide confidence bounds for the noise-perturbed and adversarially attacked OOD samples. Experiments are conducted on benchmark datasets and the performance is compared with approaches that provide asymptotic guarantees on robustness.

**Summary Of The Review:**

The proposed approach of combining OOD detection and classifier through joint training is an interesting approach. However, some parts of the paper lack clarity and thus, it is hard to evaluate the contributions. Authors are encouraged to address the comments and I will be willing to reconsider my decision.

---

> ### Author Response · Authors · 2021-11-21
> **Response to Reviewer urW9**
>
> We want to thank the reviewer for their time and for offering constructive criticism on our paper.
>
> Addressing all concerns:
>
> “*1. In table 1, metrics are not clear until section 3*.”,”*Figure 1 cannot be interpreted from the caption. It is not clear how the equations in the figure map to equations in the draft*.”\
> We have modified the captions to refer to the corresponding equations and relevant sections. We have also fixed a small typo in one of the equations shown in Figure 1.
>
> “*In the experiments section, "semi-joint training" requires more details*”\
> Note that the details of semi-joint training can be found in Appendix D and the loss function is given in Eq. (7). We have also updated Section 4.2 in order to clarify the fact that the binary discrimnator remains fixed during semi-joint training. If the reviewer has further suggestions on what should be clarified, we greatly appreciate the feedback.
>
> “*2. Authors claim to achieve better guarantees than the existing approaches (e.g., CCU, GOOD)*. *However, it is not clear why Theorem 1 entails a tighter bound than Lemma 1* (*Hein 2019*)”\
> There seems to be a slight misunderstanding as to what our Theorem 1 and Lemma 1 (Hein 2019) state. Lemma 1 (Hein 2019) states that there exists an $\alpha_0>0$ such that the set $\lbrace \alpha x | \alpha \in [\alpha_0,\infty]\rbrace$ is fully contained in one linear region (one of the convex polytopes on which the ReLU network is an affine function for every output unit). This result is then used in (Hein, 2019) to show that ReLU networks are asymptotically overconfident (the estimated conditional distribution $\hat{p}(y|x)$ converges for one class to 1 as one moves to infinity), which is obviously an undesired property.
>
> We show in Theorem 1 exactly the opposite result. Namely, our final estimate of the conditional distribution (Equation 2) over the classes $\hat{p}(y|x)$ tends to a uniform distribution over the classes as $x$ moves to infinity, which can be interpreted as the fact that the classifier states correctly that it becomes completely uncertain in its decision as it is operating far away from the training data. We achieve this result by explicitly constraining the binary discriminator to have strictly negative weights in the final layer (which has no negative impact on its performance).
> The guarantee in Theorem 1 is different from and in addition to the guarantees that Eq. (3) provides via IBP. We point out this distinction in the first paragraph of Section 3. Also note that
> OE, GOOD, ACET and ATOM do not have a guarantee that they avoid asymptotic overconfidence and in fact we show in Appendix A empirically, that all these methods also get asymptotically overconfident except GOOD. CCU is the only method which also has the guarantee that it avoids asymptotic overconfidence but CCU is not at all adversarially robust wrt to the $l_\infty$-threat model.

---

> > ### Author Response · Authors · 2021-11-21
> > **Response to Reviewer urW9 (cont.)**
> >
> > “*3. The bounds in Eq 5 appear trivial from the computations of W_{+}, and W_{-}. How do these bounds affect the tightness of the bounds in Eq 6*”\
> > In Eq. (3) we simply restate the propagation of the interval bounds from layer to layer of interval bound propagation (IBP) in order to make the paper self-contained. As discussed in the first paragraph on page 2, despite its simplicity IBP is known to produce SOTA results for certified adversarial robustness when one optimizes the bounds during training. We can see that, in practice, our bounds are remarkably tight by looking at the AAUC and GAUC in Table 2. The AAUC is an upper bound on the worst-case AUC (computed using adversarial attacks) while the GAUC is a lower bound (computed using IBP). For ProoD, these metrics are very tight with a maximal gap of less than 1% in all of Table 2. That means that even using far more sophisticated verification techniques one could only marginally improve the strength of the guarantees (also see the discussion in the first paragraph of the "Results'' section).
> >
> > “*4. I would expect the joint training would improve the performance of both OOD detection and the classifier. However, as shown in table 2, outlier exposure, which is not the best state of the art, still performs better. What are the authors' comments on this*”\
> > There is no reason to expect that our semi-joint training procedure should improve the clean AUCs beyond what OE provides. According to all prior literature one would rather expect that  certifiable methods come at a significant cost in clean performance, which is the main reason why they are not being used widely in applications.
> > Despite this, when considering all datasets (and our alternate training out-distribution 80M Tiny Images in Table 5), it is clear that ProoD performs about as well or only slightly worse than OE in terms of clean AUCs (and identically in clean accuracy) but has additionally provable guarantees for adversarially robust OOD detection. This shows that one can obtain provable guarantees at (almost) no cost in clean performance  - a finding that is highly novel and unexpected.
> >
> > “*5. Authors are encouraged to compare with the more recent OOD detection approaches*”\
> > We agree that it is always good to compare to additional baselines.
> > However, the method of Lee et al,  "Training confidence-calibrated classifiers for detecting out-of-distribution samples", ICLR 2018,  has already been shown to yield inferior performance when one does not select hyperparameters for each test out-distribution (see Meinke et al, "Towards neural networks that provably know when they don't know", ICLR 2020).
> > We added the pretrained model  from Lie et. al.'s "Energy-based out-of-distribution detection", NeurIPS 2020 (which uses 80M Tiny Images as out-distribution) as a baseline toTable 5. The EB model does not consistently outperform OE and offers neither guaranteed nor empirical adversarial robustness on the out-distribution (but also does not claim to have this).

---

### Author Response · Authors · 2021-11-21
**Summary of Contributions**

Dear Reviewers,

We want to thank all of you for taking the time to read our paper and for offering constructive criticism. We would also like to highlight  the main goal of our paper. We believe that reliable out-of-distribution detection is crucial for the safe deployment of neural networks in safety-critical applications. Certifiably adversarially robust OOD detection is one important step in that direction but practical adoption of such methods will be slow if these methods incur significant losses in clean accuracy, clean OOD detection performance or require very expensive or unstable training.

With ProoD, we demonstrate that it is indeed possible to provide certifiably adversarially robust OOD detection at no loss in clean accuracy and marginal (if any) loss in clean OOD detection performance - even on high resolution images. As Reviewer U32o correctly points out, such results are unprecedented in the traditional setting of adversarially robust models on the in-distribution (even more so for certifiably adversarially robust models).

On top of this, our method additionally provably solves the issue of asymptotic overconfidence that all ReLU classifiers have faced until now as shown in (Hein et al, 2019). The required architectural change is minor and easy to implement and thus poses few barriers for practical adoption.

---

### Decision · Program_Chairs · 2022-01-20

**Decision:**

Reject

**Comment:**

This paper aims for detecting not only clean OOD data, but also their adversarially manipulated ones. The authors propose a method for this goal, with no/marginal loss in clean test accuracy (say, Acc) and clean OOD detection accuracy (say, AUC), while existing methods for targeting the same goal suffers from low Acc and AUC. 3 reviewers are positive and 2 reviewers are negative. Reviewers and AC think that the proposed idea of merging a certified binary classifier for in-versus out-distribution with a classifier for the in-distribution task is interesting. However, AC thinks that experimental results are arguable as pointed out by reviewers. For example, in CIFAR-10, the proposed method outperforms the baseline (GOOD) with respect to Acc and AUC, but often significantly underperforms it with respect to GAUC (guaranteed AUC) or AAUC (adversarial AUC). Then, the question is which metric is more important? It is arguable to say whether Acc is more important than GAUC or AAUC. But, at least, AC thinks that AUC and AAUC (or GAUC) are equally important as adversarially manipulated OOD data is nothing but another OOD data made from the original clean OOD data. Hence, the superiority of the proposed method over the baseline is arguable in the experiments, and AC tends to suggest rejection.

ps ... AC is also a bit skeptical on the motivation of this paper. What is the value of obtaining "guaranteed AUC"? It is not the "real/true" worst case OOD performance, as it varies with respect to the tested clean OOD data. Namely, it is the worst case OOD performance just in a certain "subset" of OOD data, i.e., adversarially manipulated OOD data made from a certain clean OOD data. Hence, AC is curious about what is the value of establishing such a "partial" lower bound (rather than "true" lower bound considering all possible OOD data). AC thinks that the problem setup studied in this paper (and some previous papers) looks interesting/reasonable at the first glance, but feels somewhat artificial after a deeper look.